# Cancer of unknown primary stem-like cells model multi-organ metastasis and unveil liability to MEK inhibition

Federica Verginelli [1], Alberto Pisacane[2], Gennaro Gambardella [3,4], Antonio D'Ambrosio[1], Ermes Candiello[1], Marco Ferrio[1], Mara Panero[2], Laura Casorzo[2], Silvia Benvenuti[5], Eliano Cascardi[2,6], Rebecca Senetta[2], Elena Geuna[7], Andrea Ballabio [3,8,9], Filippo Montemurro [7], Anna Sapino [2,6], Paolo M. Comoglio[5,10] & Carla Boccaccio [1,11 ✉]

Cancers of unknown primary (CUPs), featuring metastatic dissemination in the absence of a primary tumor, are a biological enigma and a fatal disease. We propose that CUPs are a distinct, yet unrecognized, pathological entity originating from stem-like cells endowed with peculiar and shared properties. These cells can be isolated in vitro (agnospheres) and propagated in vivo by serial transplantation, displaying high tumorigenicity. After subcutaneous engraftment, agnospheres recapitulate the CUP phenotype, by spontaneously and quickly disseminating, and forming widespread established metastases. Regardless of different genetic backgrounds, agnospheres invariably display cell-autonomous proliferation and self-renewal, mostly relying on unrestrained activation of the MAP kinase/MYC axis, which confers sensitivity to MEK inhibitors in vitro and in vivo. Such sensitivity is associated with a transcriptomic signature predicting that more than 70% of CUP patients could be eligible to MEK inhibition. These data shed light on CUP biology and unveil an opportunity for therapeutic intervention.

[1] Laboratory of Cancer Stem Cell Research, Candiolo Cancer Institute, FPO-IRCCS, Candiolo, Turin, Italy. [2] Unit of Pathology, Candiolo Cancer Institute, FPO-IRCCS, Candiolo, Turin, Italy. [3] Telethon Institute of Genetics and Medicine (TIGEM), Pozzuoli, Naples, Italy. [4] University of Naples Federico II, Department of Chemical Materials and Industrial Engineering, Naples, Italy. [5] Laboratory of Exploratory Research and Molecular Cancer Therapy, Candiolo Cancer Institute, FPO-IRCCS, Candiolo, Turin, Italy. [6] Department of Medical Sciences, University of Turin Medical School, Turin, Italy. [7] Multidisciplinary Oncology Outpatient Clinic, Candiolo Cancer Institute, FPO-IRCCS, Candiolo, Turin, Italy. [8] University of Naples Federico II, Department of Medical and Translation Science, Naples, Italy. [9] Jan and Dan Duncan Neurological Research Institute, Texas Children Hospital, Houston, TX, USA. [10] IFOM, FIRC Institute of Molecular Oncology, Milan, Italy. [11] Department of Oncology, University of Turin Medical School, Candiolo, Turin, Italy. ✉email: carla.boccaccio@ircc.it

Cancer of unknown primary (CUP) is the diagnosis received by patients that present multiple metastases in the absence of a primary tumor anatomically or histologically recognizable through a standardized work-up that includes thorough body imaging and tissue immunohistochemistry[1–4]. Although implying some uncertainties, the definition of CUP applies to 1–2% of all malignancies and features a dismal median survival (<1 year). The clinical course is, however, less aggressive in a subset of cases (15–20%) displaying features reminiscent of some specific tissue of origin, such as the neuroendocrine carcinoma[2,4].

So far, CUPs were investigated with two main pragmatic clinical aims, such as to uncover the molecular or epigenetic signature of a putative "tissue of origin", and treat each CUP as a high-grade metastatic tumor of that tissue or organ[5–7], or to identify mutated cancer genes and inform personalized targeted therapies[8,9]. However, apart from a few exceptions, prediction of a putative origin and application of tailored therapies negligibly extended the overall survival of CUP patients, which remains among the poorest in oncology[3,4,10]. To make progress, it seems critical to adopt an alternative approach, and investigate CUPs as a group of tumors sharing the common ability to disseminate in a way that is (i) early and rapidly progressing; (ii) unrestrained by those barriers in the tissue of origin that facilitate the growth of a detectable primary mass; and (iii) associated with an early cell differentiation block, which precludes formation of recognizable tissues.

In this study, we isolated a panel of human CUP-initiating stem-like cells, named "agnospheres", able to reproduce a faithful disease model that features early cell dissemination and formation of established multi-organ metastases. We show that agnosphere properties are cell autonomous and converge on constitutive activation of the proliferative MAP kinase pathway, sustaining expression, and activity of the *MYC* proto-oncogene. MEK inhibition with trametinib causes agnosphere cell death, and necrosis of experimental CUP tumors, without inducing negative feedback mechanisms. Importantly, we also show that response to trametinib is foreseen by an originally set up gene expression signature, which, applied to patients' tissues, predicts sensitivity in a high percentage of CUPs, suggesting that common disease molecular mechanisms are amply shared.

## Results

**Agnospheres display the genetic makeup of original tumors.** In a cohort of 61 early metastatic cancer patients (i.e., that presented at first diagnostic imaging assessment without an evident primary tumor), enrolled at our institution, 27 were diagnosed as CUPs through a rigorous ad excludendum protocol that ruled out the presence of a primary lesion or a defined tissue of origin[2,4]. In the CUP cohort, fresh human specimens for biological studies could be obtained only in eight cases (29%), including five biopsies and three surgeries (Supplementary Tables 1 and 2). All eight samples were transplanted in immunocompromised mice (patient-derived xenografts, PDX), resulting in 6/8 engraftments (75% success rate). Interestingly, CUP samples that did not engraft belonged to a patients' subset with prolonged survival: AGN47 (OS > 40 m) and AGN913 (OS > 84 m). From PDX, or fresh human samples whenever possible, we attempted spheroid cultures of stem-like cells, named "agnospheres", with a 75% of successful rate (Fig. 1a and Supplementary Table 2). Briefly, five out of six patients that provided agnospheres (AGN901, AGN906, AGN43, AGN67, and AGN914) displayed the typical aggressive CUP presentation, featuring (i) multiple metastases; (ii) the histological aspect of poorly differentiated carcinomas lacking expression of markers associated with specific organs or tissues, and (iii) rapidly lethal

clinical outcome (average overall survival: 11.5 months; Supplementary Table 1). Agnospheres derived from the above patients are hereafter indicated as AS901, AS906, AS43, AS67, and AS914, respectively. Patient AGN47 presented with multiple metastases but, unlike the above cases, displayed signs of differentiation, leading to the diagnosis of "neuroendocrine CUP" and a better prognosis. The corresponding agnosphere is henceforth indicated as N-AS47.

We performed the genetic characterization of agnospheres and the corresponding available human tissues (hereafter indicated as "original tissues"), by whole-exome sequencing (Supplementary Table 3): AS901 and AS67 resulted hypermutated, consistently with the presence of a *POLE* mutation in AS901, and *POLE* and *POLQ* mutations in AS67; AS906, AS43, and AS914 harbored different combinations of cancer-associated genes. In contrast, N-AS47 harbored no mutation in known oncogenes or tumor suppressor genes (TSG; Supplementary Table 3). Copy-number variation analysis of the two most commonly altered tumor suppressor genes unveiled heterozygous loss of *TP3* accompanied by *TP53* hemizygous mutation in AS43, and heterozygous *PTEN* loss in both AS914 and N-AS47 (Fig. 1b and Supplementary Table 3). The variant allele frequency is consistent with heterozygosity (oncogenes) or hemizygosity (TSG) in agnospheres (Supplementary Table 3). This supports the pathogenic meaning of the genetic alterations and suggests that agnospheres are monoclonal cell populations, at least relatively to driving genetic lesions. Interestingly, we recently showed that multiple synchronous metastases, sampled at warm autopsy of patient AGN43, are highly genetically related and, in particular, share all the driver gene mutations[11], thus indicating that agnospheres, although derived from a single metastatic lesion, can be representative of all patient's metastases.

Conventional and spectral karyotypic analysis showed that AS901, consistently with its hypermutated state, did not display evident aberrations of chromosome number and structure (Fig. 1c). In contrast, AS906, AS43, AS914, and AS67 (although hypermutated), exhibited slightly increased ploidy and multiple numerical and/or structural chromosome aberrations, while N-AS47 showed multiple numerical but few structural aberrations (Fig. 1c, d and Supplementary Notes). Although the sample size is too small to generalize conclusions, the overall genetic analysis indicated mutational and karyotypic heterogeneity among agnospheres, consistently with previous observations in 200 CUP patients[8].

**Agnospheres are enriched in self-sustaining stem-like cells.** Agnospheres were isolated and propagated in suspension, in highly stringent stem culture conditions[12]. Surprisingly, all agnospheres were capable of self-renewing and long-term propagating at clonal density in the complete absence of any exogenous growth factor, including EGF or FGF, usually required for in vitro isolation and long-term maintenance of the stem phenotype in cancer stem cells from highly aggressive tumors[12–15]. Indeed, in our previous studies, cancer stem cells derived from glioblastomas or colorectal cancer metastases could be in vitro long-term propagated in the absence of exogenous growth factors only in rare and highly aggressive cases, mostly featuring constitutive activation of the growth factor receptor/RAS pathway[15–17]. Among these, for comparison with agnospheres, we choose, as representative carcinoma-derived tumorspheres, colorectospheres mCRC729 and mCRC0155, both generated from colorectal cancer liver metastases, and harboring *IGF-2* gene locus amplification or *KRAS* mutation, respectively[17]. Moreover, we newly generated a growth factor-independent melanosphere (mMS321), from a visceral metastasis with unequivocal melanocytic differentiation in

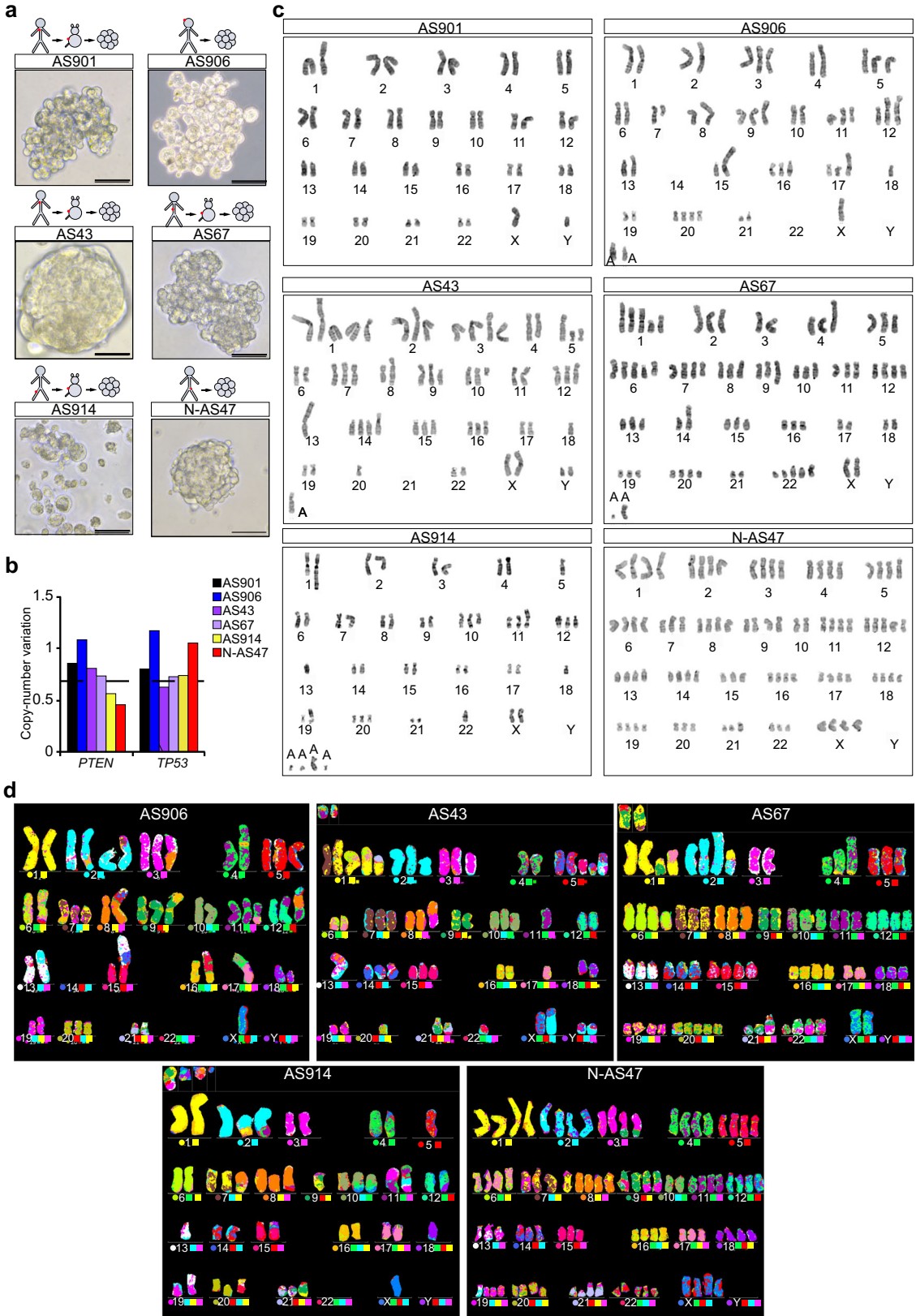

**Fig. 1 Agnospheres can be isolated from cancers of unknown primary and display the genetic makeup of the original tumors. a** Morphological appearance of agnospheres derived from CUP tissues propagated in mice or directly from human tissues. **b** *PTEN* and *TP53* copy-number variations measured by qPCR in gDNA from agnospheres. Value = 1 indicates biallelic content; values < 0.7 (dotted line) indicate allelic loss, ($n = 2$ independent experiments with similar results were obtained). **c** Representative G-banded karyotypes of agnospheres. A: marker chromosome. **d** Representative images of spectral karyotypic analysis (M-FISH) on metaphases of: AS906 ($n = 16$) showing a near-diploid karyotype; AS43 ($n = 9$), containing two major clones with different karyotypes (near-diploid and hypotriploid); AS67 ($n = 20$) with hypertriploid modal karyotype; AS914 ($n = 20$) showing a near-diploid modal karyotype; and N-AS47 ($n = 13$), including three major clones (near-diploid, near-triploid, and near-tetraploid).

the absence of a detectable primary skin lesion, and harboring a *BRAF* mutation (Supplementary Table 1). Independently of exogenous growth factors, all agnospheres displayed the ability to self-sustain the expression of high levels of transcription factors (TFs) known to be major regulators of self-renewal and stem identity in normal and neoplastic tissues, such as Polycomb repressors[18] and reprogramming TFs[19,20]. Specifically, Polycomb repressor EZH2 and BMI1 were highly expressed in the majority of agnosphere cells (Fig. 2a–c and Supplementary Fig. 1a). Among reprogramming TFs, prominent expression of *MYC* gene products, in the absence of gene amplifications (Supplementary Fig. 1b), was observed in all agnospheres (Fig. 2a–c). Co-expression of MYC with OCT4, SOX2, and KLF4 was observed only in a fraction of agnosphere cells, likely marking a subpopulation with enhanced stem traits (Fig. 2b and Supplementary Fig. 1a). In tumorspheres, Polycomb and reprogramming TFs were expressed as well, although collectively at a lesser extent, in particular in colorectospheres (Fig. 2c and Supplementary Fig. 1c). Interestingly, YAP/TAZ, recently implied in reprogramming of differentiated cells into stem cells[21], was highly expressed in all aggressive CUPs, but undetectable in N-AS47 (Fig. 2c).

As the activity of the above TFs is known to drive genetic programs sustaining the embryonic stem (ES) status, in the overall agnosphere transcriptome we investigated the expression of an "ES cell signature", based on transcription of 13 gene sets collected in four main groups, including (i) the Polycomb and (ii) the MYC target groups, (iii) a group encompassing targets of NANOG, OCT4, and SOX2, and (iv) a group of genes specifically expressed in cultured human ES cells (Fig. 2d and Supplementary Data 1)[22]. By direct comparison, agnospheres and tumorspheres displayed global levels of Polycomb and MYC target genes overall similar and comparable to those of ES cells (Fig. 2d and Supplementary Data 1), although N-AS47 and colorectospheres displayed slighter repression of Polycomb targets. Notably, while, in ES cells, maintenance of the transcriptional program sustaining the ES status requires exogenous factors[19,20], in agnospheres most of the same program is driven in a fully cell-autonomous manner. As the "ES cell signature" was previously shown to be enriched in poorly differentiated aggressive human tumors, and directly correlated with breast cancer tumor grade[22] we investigated its expression in our panel of CUP tissues (including four matched with agnospheres: AGN901, 906, 43, and 47), and, for comparison, in a previously reported panel of breast cancer transcriptomes[22]. CUP tissues displayed an "ES cell signature" globally weaker compared with agnospheres, as expected, but significantly more enriched compared with grade 3 breast cancer tissues (Fig. 2d and Supplementary Data 1), known to contain an abundant fraction of cells with stem properties[13]. Overall transcriptomic data suggest that CUP tissues contain a high proportion of cells that retain stem traits, consistently with their poorly differentiated histology.

Given the interconnection between stem phenotype, metastatic ability, and epithelial–mesenchymal transition (EMT)[23], we assessed protein expression of EMT core TFs ZEB1 and SNAI2, epithelial markers E-Cadherin and EpCAM, and mesenchymal markers Vimentin and CD44 (Fig. 2c). Moreover, we analyzed the full panel of core and accessory EMT TFs in the agnosphere transcriptome (Supplementary Fig. 1d)[24]. Agnospheres derived from CUP adenocarcinomas (AS901, AS43, and AS67, Supplementary Table 1) and neuroendocrine carcinoma (N-AS47), which maintained in culture an epithelioid phenotype (Fig. 1a), did not display EMT TF or marker expression (Fig. 2c). In contrast, AS906 and AS914, which derived from carcinomas with sarcomatoid features (Supplementary Table 1) and formed loose spheroids in culture (Fig. 1a), expressed ZEB1, SNAI2, CD44, and Vimentin, but negligible levels of EpCAM and E-Cadherin

(CDH1; Fig. 2c and Supplementary Fig. 1e). Accordingly, transcriptomic analysis showed global higher EMT TF expression in AS906 and AS914, while the maximum levels were observed in mMS321 (Supplementary Fig. 1e). This analysis showed also that CUP tissues globally displayed higher EMT TF expression as compared with agnospheres, possibly as result of contamination by tumor microenvironment cells. Validation by tissue immunostaining could not be performed for lack of material, leaving undetermined the real EMT extent in original CUP tissues. Altogether, these findings suggest that EMT seems constitutively activated only in agnospheres derived from CUPs displaying sarcomatoid (i.e., mesenchymal) features. In the other cases, clearly EMT does not occur spontaneously in basic culture conditions, but it could be activated by appropriate exogenous signals and play a key role during metastatic dissemination.

Finally, in the agnosphere and CUP tissue transcriptome, we also analyzed the expression of cell surface and functional stem cell markers previously used for identification and/or prospective isolation of cancer stem cells from different types of carcinoma. In both agnospheres and CUP tissues, we found significant expression of general markers, such as CD24, CD98, CD166, ITGA6, and ITGB1, with the notable exception of CD133 that was rarely expressed (Supplementary Fig. 1e)[25]. Interestingly, the MYC target gene and "don't eat me" immunosuppressive signal CD47 (refs. [26,27]) was broadly expressed (Supplementary Fig. 1e). The overall heterogeneity of cell surface marker expression prevented the design of a common strategy for future prospective isolation of CUP cancer stem cells and further characterization of agnosphere subpopulations.

Concerning functional properties, agnospheres kept in the absence of exogenous growth factors displayed an in vitro estimated fraction of clonogenic cells ranging between 15 and 44% in AS901, AS906, AS43, and AS67, which decreased to 4% in AS914 and to a mere 0.09% in N-AS47 (Fig. 2e). The clonogenic frequency inversely correlated with the population doubling time, which was relatively short in the highly self-renewing agnospheres (2 days in AS901, AS906, and AS67, and 3.5 days in AS43) and increased to ~5 and 10 days in AS914 and N-AS47, respectively.

Overall these data indicate that agnospheres are enriched in cells with transcriptional traits typical of ES cells, mirrored by functional clonogenic properties, which are fully self-sustained in the absence of exogenous growth factors. Such transcriptional features are present also in CUP tissues, attesting to the faithful phenotypic correspondence between the original tumors and the agnospheres, and indicating that CUP tissues are highly enriched in stem-like cells as well.

**Agnospheres generate phenocopies of the original tumors and contain a high percentage of tumor-initiating cells.** Next, to assess the agnosphere tumorigenic potential, we subcutaneously transplanted $10^5$ agnosphere cells in immunocompromised mice (spheropatients). At the injection site (IS), tumors formed in all mice, within 20 days in the case of all aggressive agnospheres, and only after 3 months in the case of N-AS47, reflecting the slower progression of the corresponding human tumor (Supplementary Table 1) and the long in vitro cell population doubling time. Importantly, agnospheres regenerated IS-tumors indistinguishable from the original human tumors by histology and expression of immunohistochemical markers (Fig. 3a and Supplementary Fig. 2a), a defining prerogative of cancer stem cells[28]. Interestingly, agnospheres retained in culture the same tissue marker expression, thus behaving as tumoroids (Fig. 3a and Supplementary Fig. 2a)[29]. This property highlights the agnosphere ability not only to proliferate, but also to recapitulate pseudodifferentiative programs

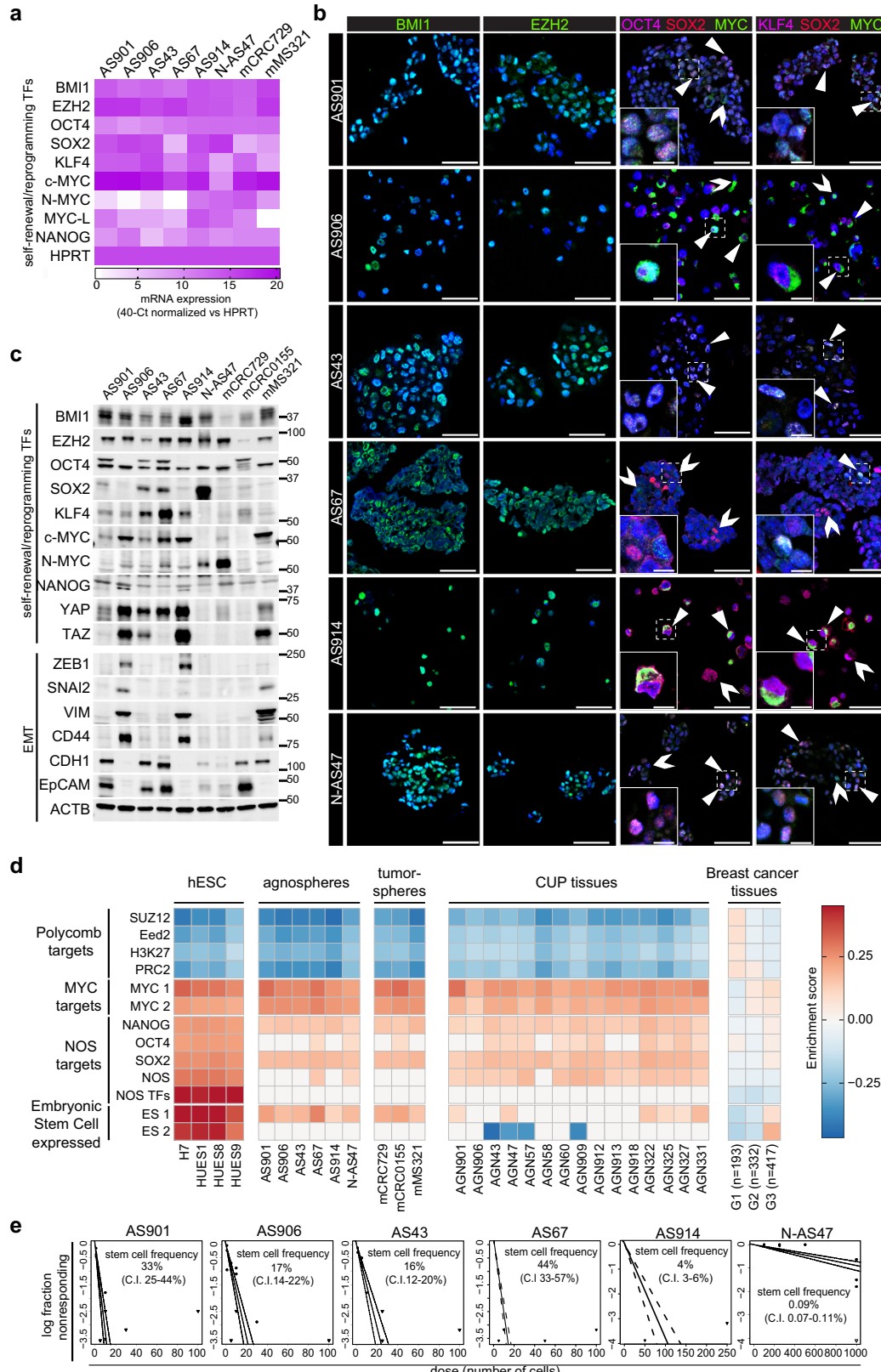

independently of microenvironmental cues. Consistently with the phenotypic and transcriptional correspondence between agnospheres and original tumors (Fig. 2d), strong and widespread protein expression of Polycomb repressors was displayed by both original and IS-tumors (Fig. 3b and Supplementary Fig. 2b), as well as by agnospheres (Fig. 2b), suggesting that agnosphere

culture conditions did not significantly modify genetic programs and phenotypic traits inherited by original tissues.

In order to evaluate the tumor-initiating potential, we challenged all agnospheres by stringent in vivo limiting dilution serial transplantation experiments up to three passages. Remarkably, in all cases but N-AS47, as few as ten agnosphere cells could

**Fig. 2 Agnospheres are enriched in stem-like cells that self-sustain their long-term propagation. a** Heatmap showing gene expression levels of selected transcription factors (TFs), quantified by qRT-PCR in agnospheres and tumorspheres derived from metastatic colorectal cancer (mCRC729) or melanoma (mMS321). **b** Representative immunofluorescent stainings of stem cell markers in agnospheres. Arrowheads: marker co-expression; open arrows: cells with single marker expression (n = 3 independent stainings of agnospheres with similar results were obtained). Scale bar, 50 μm. Inset: magnification of dotted area. Scale bar, 10 μm. **c** Representative western blot analysis of Polycomb repressor factors, reprogramming and EMT TFs, and epithelial and mesenchymal markers in agnospheres and tumorspheres mCRC729, mCRC0155, and mMS321 (n = 3 independent experiments with similar results were obtained, molecular weights are expressed in kDa). **d** Heatmap showing the gene set enrichment analysis of the 13 gene sets included in the embryonic stem cell signature[22], performed on the transcriptome of human embryonic stem cell lines (hESC), agnospheres, tumorspheres from metastatic cancers, CUP original tissues, and a panel of breast cancer tissues, pooled by grade (the average enrichment score is shown for each grade). Only significant enrichment scores with a false discovery rate < 10% are shown (Supplementary Data 1). NOS: NANOG, OCT4, and SOX2. **e** In vitro limiting dilution sphere-forming assay. For each agnosphere, plots generated by the ELDA software are shown, reporting the estimated stem cell frequency (percentage of clonogenic cells) with confidence intervals (C.I.; AS901, AS43, AS67, and AS914: n = 3; AS906 and N-AS47: n = 4 independent experiments).

generate tumors after subcutaneous transplantation, determining an estimated tumor-initiating cell (TIC) frequency ranging between 0.3 and 25% at the first passage (P0), and increasing up to 50% in AS901 and AS906 at the third passage (P2; Fig. 3c). Not surprisingly, N-AS47 displayed a relatively low TIC frequency (0,002%), consistently with its overall in vitro biological properties and modest clinical aggressiveness (Fig. 3c). Notably, TIC frequencies in aggressive agnospheres are comparable to the TIC frequency (8.5%) that we measured in melanosphere mMS321 after subcutaneous transplantation (Supplementary Fig. 2c). Melanoma is known as the tumor displaying the highest TIC frequency, also because the subcutis provides an orthotopic environment favoring cell viability and spread[30]. In contrast, tumorspheres derived from other aggressive cancers, when assessed in the same in vivo conditions (subcutaneous transplantation in NOD/SCID mice), display TIC frequencies 1–2 log lower than agnospheres: colospheres up to 0.2% (ref. [31]), breast cancer mammospheres up to 0.1% (ref. [32]), and glioblastoma neurospheres up to 0.1% (ref. [33]).

Altogether these findings highlight the agnosphere ability to recapitulate the original CUP tumors, either by reproducing their histological and pseudodifferentiative features, or by mirroring their clinical aggressiveness in TIC content.

**Agnospheres reproduce the multi-organ metastatic pattern of CUP patients.** We then assessed metastatic ability, by transplanting subcutaneously luciferase-labeled agnosphere cells from four representative cases (AS901, AS906, AS43, and N-AS47). After removing IS-tumors, to prevent both overgrowth and interference with weaker luminescence from metastatic sites, we longitudinally monitored spheropatients by in vivo imaging. Increasing luminescent signals were detected in vivo at multiple sites, and further visualized in the explanted organs (Fig. 4a–c). Histopathological and immunohistochemical analysis showed that agnospheres generate metastases phenocopying the original and the IS-tumor (Fig. 4d and Supplementary Fig. 3a–c), indicating that agnospheres retain specific pseudodifferentiative programs rather independent from exogenous signals or tissue contexts, and can adapt to grow in different environments. In spheropatients transplanted with the aggressive agnospheres (AS901, AS906, and AS43), the metastatic burden lead to the clinical endpoint within 4 months, and the overall postmortem analysis showed multiple metastatic sites in ~75% of mice (Fig. 4e). In N-AS47 spheropatients, metastatic growth was not fatal, mirroring the patient's clinical course (characterized by a long survival >40 months). However, at the experimental end-point (7 months), widespread micrometastases and disseminated single cancer cells were detected in 100% of mice, including a remarkable fraction (54% of all mice), where the IS-tumor did not grow (Fig. 4e).

Among the metastatic sites, we found axillary lymph nodes (Fig. 4f), considered as loco-regional lymph nodes invaded by lymphogenous spread, which often occurs in CUP patients as well (Supplementary Table 1)[4]. The other organs colonized in spheropatients were likely reached by hematogenous spreading, with homing occurring not only in the primary capillary district downstream venous dissemination (lung), but, often, downstream systemic arterial circulation as well. The frequent colonization of connective tissues, including peripancreatic fibroadipose tissue, gonadal adipose tissue, kidney capsule, and the occasional homing between myocardiocytes, suggested a predilection for the connective/mesodermal soil, where micrometastases and intravascular cancerous emboli could be often detected (Fig. 4d and Supplementary Fig. 3a–c). This, again, is consistent with the CUP metastatic pattern, which can include uncommon sites, such as subcutaneous connective tissues and muscles, as observed in several patients, including AGN906, AGN43, AGN914, AGN909, AGN322, and AGN325 in our cohort (Supplementary Table 1).

Strikingly, in CUP spheropatients, metastatic cells efficiently disseminated as early as within 10 days after injection, as observed in organs explanted from 4/4 mice transplanted with AS43. Dissemination occurred well before the IS-tumor became palpable, suggesting that cells can initiate the process without prior local expansion and, likely, without induction of a pre-metastatic niche from an established tumor. Immunostaining of explanted organs showed single disseminated tumor cells, micrometastases and vascular emboli at multiple sites, mostly in the lung and connective tissues around the organs, i.e., the same sites where macrometastases develop (Fig. 4g). Interestingly, single disseminated cells were identified by pan-cytokeratin immunostaining, suggesting that, during metastatic spread, these cells retain their epithelioid phenotype, or reside in a partial EMT state.

Such a rapid and widespread, both lymphogenous and hematogenous, metastatization is rather uncommon when TICs or organoids from other aggressive carcinomas are transplanted subcutaneously (i.e., ectopically), in contrast to neural crest-derived melanoma cells, which, injected orthotopically under the skin, can metastasize[34]. Consistently, we could not observe metastatic dissemination by subcutaneously transplanting mice with colosphere mCRC729, which, as mentioned above, derives from a colorectal cancer metastasis and shares some properties with agnospheres (Fig. 2a–d), in particular cell-autonomous propagation. Indeed, after mCRC729 injection, IS-tumors formed in 100% of cases, being in some cases locally invasive, but always unable to form detectable metastases during 7-month in vivo longitudinal monitoring and ex vivo organ imaging (Fig. 4e and Supplementary Fig. 3d, e).

Overall, these findings attest that agnospheres, in particular those from aggressive CUPs, retain a high spontaneous and multi-organ metastatic ability, which recapitulates the CUP

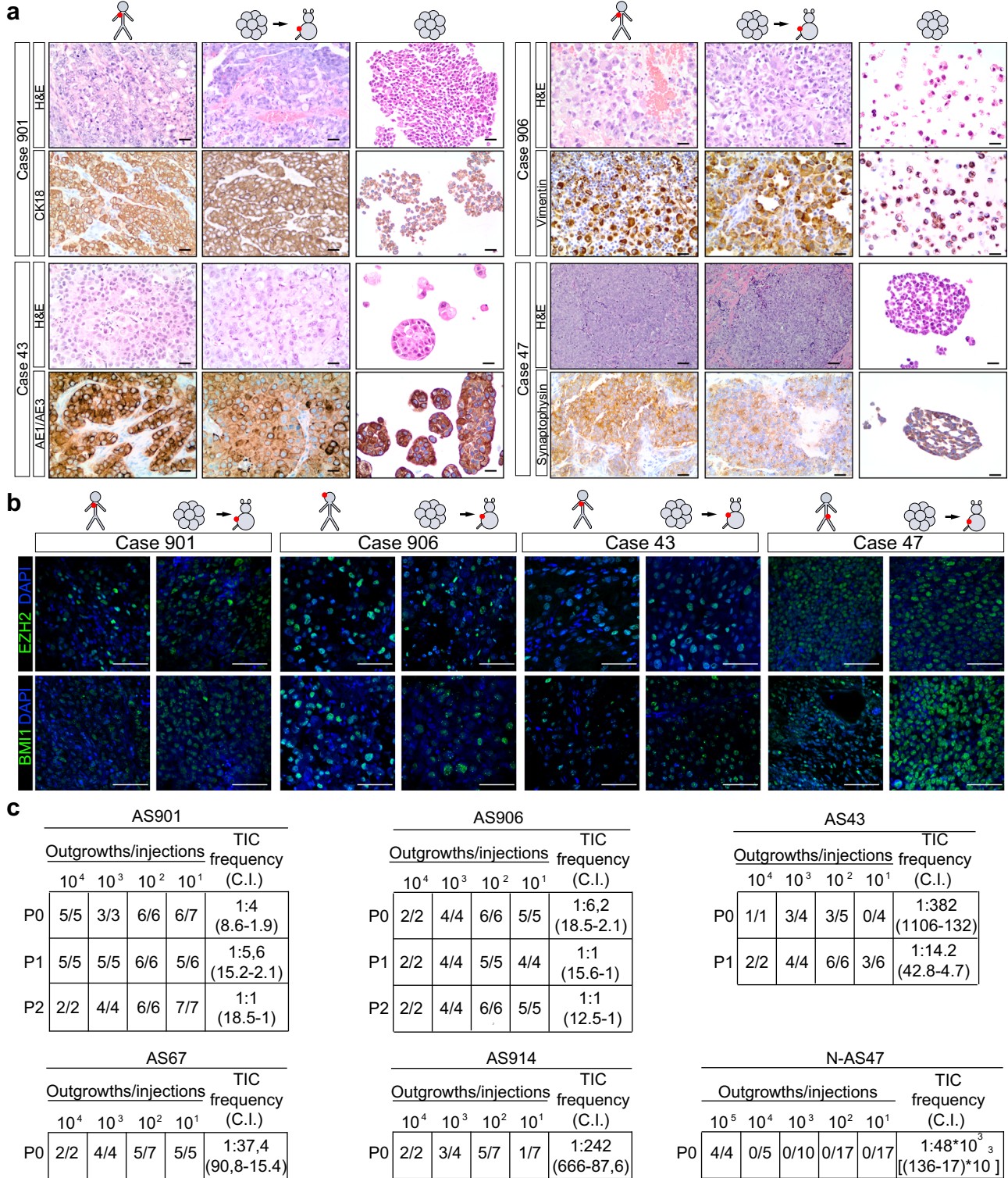

**Fig. 3 Agnospheres generate phenocopies of the original tumors and contain a high percentage of tumor-initiating cells. a** Histopathological features (hematoxylin and eosin staining, H&E) and immunohistochemistry with antibodies used for CUP diagnosis, in patient metastases (left columns), tumors formed by agnospheres at the subcutaneous injection site (middle columns), and cultured agnospheres (right columns). Scale bar, 50 µm. A representative experiment is shown (n = 3 independent stainings of agnospheres with similar results were obtained). **b** Immunofluorescent stainings of stem cell markers in patient metastases (left columns) and tumors formed by the corresponding agnospheres at the subcutaneous injection site (right columns). Nuclei were counterstained with DAPI. Scale bar, 50 µm. A representative experiment is shown (n = 3 independent stainings of agnospheres with similar results were obtained). **c** In vivo limiting dilution assay. The indicated numbers of agnosphere cells ($10^5$–$10^1$) were transplanted subcutis into immunocompromised mice (P0) and, where indicated, agnospheres were re-derived in culture and transplanted for a second (P1) or a third (P2) passage. TIC tumor-initiating cell frequency, C.I. confidence interval.

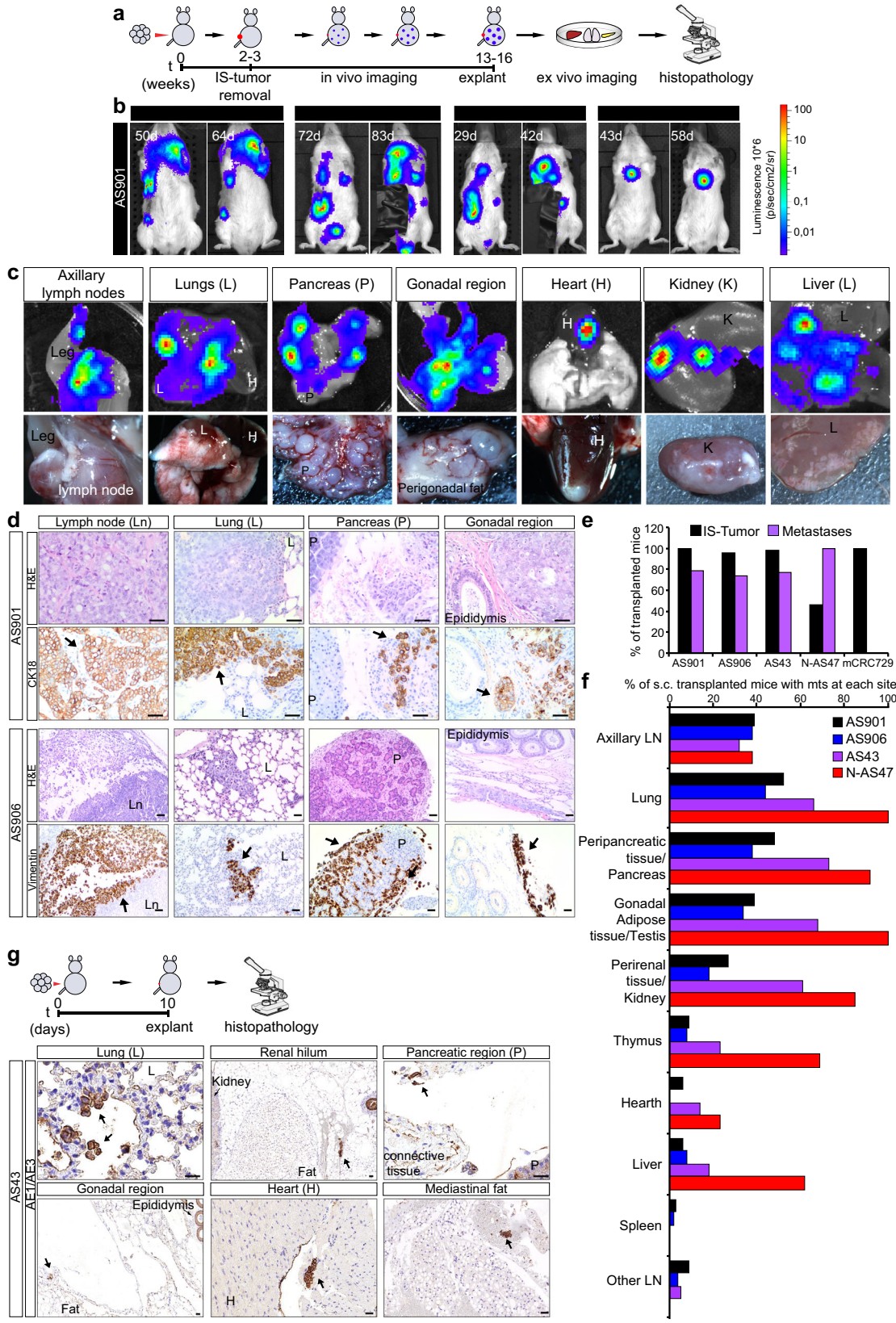

clinical presentation, and may represent a convenient tool to explore every stage of the metastatic cascade.

**Agnospheres are sensitive to MEK inhibition in vitro.** As described above, all agnospheres shared the distinctive ability to

proliferate and self-renew in the absence of any exogenous growth factor. Moreover, agnospheres, including the slowly proliferating N-AS47, were insensitive to an ample panel of exogenously supplied growth factors, with the exception of AS67, which showed a moderate proliferative response to EGF, NRG1, and FGF (Supplementary Fig. 4a). Agnosphere proliferative autonomy

**Fig. 4 Subcutaneously transplanted agnospheres reproduce the early and multi-organ metastatic pattern of CUP patients. a** Outline of longitudinal in vivo monitoring of spheropatients transplanted subcutis with luciferase-labeled agnospheres, followed by endpoint ex vivo organ imaging and histopathological analysis. IS-tumor injection site tumor. **b** Representative in vivo images of mice transplanted with AS901, showing increasing bioluminescent signals at metastatic sites at the indicated time points (d, days). Black tape was applied to mask the IS-tumor residual signal. **c** Representative macroscopic images of the most frequent metastatic sites. Top: bioluminescent signals of different explanted organs; bottom: photographs of macrometastases. **d** Histopathology (H&E) and immunohistochemistry for human CK18 or human Vimentin, revealing metastases in the indicated organs of mice transplanted with AS901 or AS906, respectively. Scale bar, 50 μm. Representative experiments are shown (at least $n = 3$ stainings in independent organs were obtained with similar results). **e** Bar graph showing the percentage of mice transplanted subcutis with agnospheres or colosphere mCRC729 that developed IS-tumors and/or metastases (AS901: $n = 33$, AS906: $n = 50$; AS43: $n = 44$; N-AS47: $n = 13$; mCRC729: $n = 14$). **f** Bar graph showing the percentage of mice subcutaneously (s.c.) transplanted with agnospheres as in **e** that displayed metastasis (mts) at each of the listed sites. **g** Immunostainings for pan-cytokeratin AE1/AE3 showing tumor cells disseminated in organs explanted 10 days after AS43 subcutaneous injection. Scale bar, 50 μm ($n = 4$ independent experiments with similar results were obtained).

could be associated with the presence of *RAS* oncogenic mutations, known to confer growth independence to tumorspheres[15], only in the case of AS906 (*HRAS*-mutated) and AS914 (*KRAS*-mutated; Supplementary Table 3). Remarkably, in all agnospheres, proliferative autonomy could be sustained by the expression of several growth factors, among which EGF-like ligands, neuregulins, HGF, and IGF-1/2 (Supplementary Fig. 4b), accompanied by expression and phosphorylation of their genetically unaltered cognate receptors (Fig. 5a), indicating the occurrence of autocrine loops. Conversely, expression and phosphorylation of the same receptors was almost undetectable in tumorspheres that relied on cell-autonomous proliferative triggers, such as BRAF (mMS321) or KRAS (mCRC0155) mutation, or amplification of the *IGF-2* genetic locus (mCRC729)[17]. These findings prompted us to try to halt agnosphere proliferation by blocking constitutively activated receptors, in particular those of the EGFR family, which seemed prominently activated in all aggressive agnospheres. Pan-EGFR family inhibition through lapatinib or afatinib[35] was assessed in three representative agnospheres: AS901 (hypermutated), AS906 (*HRAS* mutant), and AS43 (RAS pathway wild type; Supplementary Table 3). A significant proliferative arrest, without cell death induction even after a 14-day treatment, was observed only in AS43 (Supplementary Fig. 4c, d). Indeed, although abolishing EGFR phosphorylation in all agnospheres, EGFR inhibitors partially affected downstream signaling only in AS43 (Supplementary Fig. 4e). Agnosphere partial or full resistance to EGFR inhibition could be explained by the concomitant presence of the *HRAS* mutation (in AS906), as described[15], and/or other autocrine loops providing bypass mechanisms that kept MAPK and AKT pathways constitutively active (Fig. 5a and Supplementary Fig. 4e).

We therefore reasoned that the multiple mechanisms sustaining agnosphere autonomous proliferation should converge onto the MAPK pathway. As, indeed, agnospheres display high levels of constitutively phosphorylated MEK1/2, in some cases comparable to those of *BRAF*-mutated mMS321 (Fig. 5a), we challenged agnospheres with two selective, clinically approved, MEK1/2 inhibitors, endowed with different mechanisms of action: (i) trametinib, known to be the most potent MEK1/2 inhibitor, used in *BRAF*-mutated melanomas[36,37], or (ii) selumetinib, an earlier inhibitor, known to be less effective than trametinib in fully preventing MEK1/2 activation by BRAF. As assessed by dose–response experiments (Supplementary Fig. 4f), in three representative agnospheres trametinib strongly affected cell viability at nanomolar concentrations, while selumetinib was less effective even at maximal doses (Supplementary Fig. 4f), as reflected by early signaling events, such as ERK1/2 and pRB phosphorylation (Supplementary Fig. 4g). The biological effects exerted in agnospheres by the two inhibitors were comparable to those observed in *BRAF*-mutated mMS321 (Supplementary Fig. 4h). Trametinib was therefore chosen for further experiments, showing to be effective in 4/6 agnospheres, by inducing dramatic growth arrest within 4 days in AS901 and AS906, and within 10 days in

AS43 and AS67 (Fig. 5b), and concomitant apoptosis, as indicated by the appearance of the cell death effector cleaved-PARP protein (Fig. 5c). In sensitive agnospheres (AS901, AS906, AS43, and AS67), trametinib induced long-term inhibition of ERK1/2 phosphorylation, the direct MEK1/2 target, and downregulation of c-MYC, known to be tightly regulated by proliferative signals via MAPK pathway (Fig. 5c)[38,39]. c-MYC decrease (or disappearance, in the most sensitive agnospheres) was accompanied by proportional downregulation of its transcriptional target Cyclin D1, responsible for G1–S cell cycle progression via CDK4/6-mediated pRB hyperphosphorylation, which was consistently reduced as well (Fig. 5c). Stabilization of the cell cycle inhibitor p27-KIP1, known to be prevented by c-MYC, was also observed (Fig. 5c)[39]. Interestingly, in sensitive agnospheres, trametinib did not cause a rebound increased EGFR phosphorylation and expression, often observed after MEK inhibition and resulting from relief of a negative feedback from MAPK to receptors (Fig. 5c)[37]. Rather, in sensitive agnospheres, EGFR phosphorylation and expression were decreased (Fig. 5d). This suggests that agnospheres may lack this negative feedback from MAPK to receptors, resulting in an unrestrained receptor–MAPK signaling axis, which could become addictive and would eventually cause hypersensitivity to MEK1/2 inhibitors (see "Discussion"). Conversely, trametinib was biologically ineffective in N-AS47 (Fig. 5b), consistently with its inability to prevent ERK1/2 phosphorylation and downstream signaling (Fig. 5c), thereby suggesting that N-AS47 can self-sustain proliferation through pathways independent of MEK1/2, as recently shown in organoids from neuroendocrine tumors[40]. Surprisingly, in *KRAS*-mutated AS914, trametinib inhibited ERK1/2 phosphorylation but exerted a late paradoxical effect, by increasing cell proliferation (Fig. 5b, c), as described in several cell lines and tumors[37]. This effect was likely a specific outcome of MEK1/2 inhibition, as it was observed after selumetinib treatment as well (Supplementary Fig. 4i). The mechanism of such paradoxical response does not involve EGFR rebound upregulation; however, it likely contributes to keep cMYC and Cyclin D1 highly expressed independently of ERK1/2 inhibition (Fig. 5c).

**Trametinib is effective in CUP preclinical models and its efficacy is predicted in patients by a response signature.** To evaluate the therapeutic efficacy of trametinib in preclinical models, CUP tumors were generated by subcutaneous transplantation of three representative agnospheres (AS901, AS906, and AS43). After IS-tumor establishment, spheropatients were randomized and treated with a dose of trametinib (1 mg/Kg/die) previously shown to be well tolerated and effective in xenograft models harboring *BRAF* mutation (Fig. 5c)[37,41]. We chose to primarily monitor the therapeutic effect in the established IS (subcutaneous) tumor, assuming that (i) the IS-tumor is not representative of a conventional primary tumor, but rather of every metastasis generated by the agnosphere in the mouse, given the homogeneity of IS-tumors and metastases; (ii) the effect of

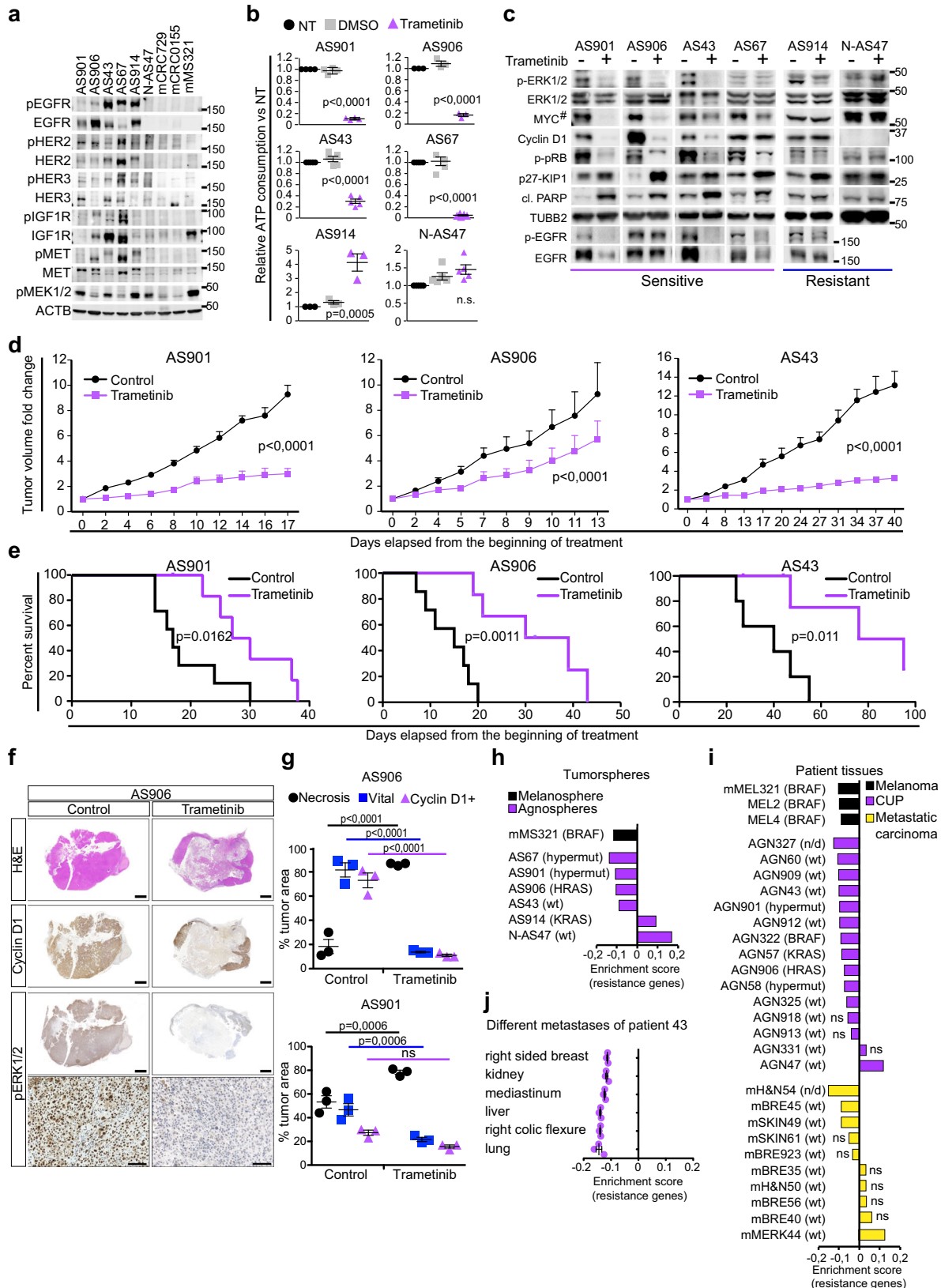

trametinib is not expected on the early dissemination step of the metastatic process, but on the growth of established metastases; (iii) the IS-tumor can be more accurately and precociously measured than metastases, which are multiple and delayed, and evolve by an unpredictable pattern that prevents reliable longitudinal quantification. The inhibitory effect of trametinib on IS-

tumor growth was statistically significant throughout the experiment (Fig. 5d). Accordingly, the time to reach the experimental endpoint (tumor volume = 1600 mm$^3$) was significantly prolonged in all mice treated with trametinib as compared with controls (Fig. 5e). Interestingly, histopathological analysis of tumors collected at the endpoint showed that treated tumors, as

**Fig. 5 MEK inhibition is effective in vitro and in CUP preclinical models, and its efficacy is reliably predicted in patients by a gene expression signature. a** Representative western blot analysis of phosphorylated (p) and total proteins in agnospheres and tumorspheres ($n = 3$ independent experiments). **b** Cell viability of agnospheres treated for 4 days (AS901 and AS906) or 10 days with 50 nM trametinib or vehicle (DMSO) normalized vs. untreated cells (NT; AS901, AS43, and AS67 $n = 4$; AS906 and AS914 $n = 3$; and N-AS47 $n = 5$ independent experiments, mean ± SEM, ANOVA, Bonferroni multicomparison test). **c** Representative western blot analysis of agnospheres treated 48 h (AS901, AS906, and AS67) or 96 h (AS43, AS914, and N-AS47) with vehicle (−) or 50 nM trametinib (+) (p- phosphorylated, cl. PARP: cleaved PARP, MYC# c-MYC in AS901, AS906, AS43, AS67, and AS914, and N-MYC in N-AS47; $n = 3$ independent experiments). **d, e** Mice were injected subcutis with agnospheres and, after tumor establishment, randomized in two treatment groups (control and trametinib, 1 mg/Kg/die; AS901, $n = 7$ mice/group; AS906, control $n = 6$ mice and trametinib $n = 7$ mice; AS43, $n = 6$ mice/group). **d** Growth curves of tumor volume fold changes vs. day 0 (beginning of treatment; mean ± SEM, two-way ANOVA). **e** Survival curve at the experimental endpoint (tumor volume = 1600 mm$^3$; Mantel–Cox test). **f** Representative H&E and immunohistochemical stainings for Cyclin D1 and phospho-ERK1/2 of whole tumor sections derived from endpoint mice transplanted with AS906 and treated with control or trametinib ($n = 3$ mice). Scale bar, 2 mm. Bottom row: magnification of pERK1/2 staining. Scale bar, 100 μm. **g** Quantification of necrotic, vital, and Cyclin D1-positive areas in whole tumor sections from mice transplanted with AS906 (top) or AS901 (bottom), expressed as percentage of total tumor area ($n = 3$ mice/group, mean ± SEM, two-way ANOVA, Sidak's multicomparison test; ns not significant). **h–j** Bar graphs of enrichment scores of the trametinib response signature in agnospheres and melanosphere mMS321 ($n = 3$ replicates) (**h**); tumor tissues from CUPs (AGN−), melanomas (MEL−), and early metastatic cancers of known origin (H&N head and neck, BRE breast, MERK merkeloma) (**i**); and different metastases of CUP patient AGN43 ($n = 3$ fragments/metastatic site, mean ± SEM) (**j**). Sensitivity to trametinib associates with a negative enrichment score of resistance genes (gene set enrichment analysis. Only enrichment scores with a false discovery rate, FDR < 10% are considered statistically significant, $p$ values and FDR values are reported in Supplementary Data 1). The presence or absence (WT) of a mutated gene associated with trametinib sensitivity (*BRAF, KRAS, HRAS,* or *NRAS*) is reported. Hypermut hypermutated, n/d not determined, ns not significant.

compared with controls, contained an ample central necrotic area surrounded by proliferating cells, still responding to trametinib with ERK1/2 inhibition (Fig. 5f, g and Supplementary Fig. 4j). This indicates that tumor volume measurement alone underestimates the trametinib biological effect, while in vitro proliferation arrest and cell death reliably predict the in vivo tumor tissue response to trametinib.

To attempt an evaluation of trametinib effects on spontaneous metastases, we analyzed serial sections of whole lungs (the site most frequently colonized in spheropatients), collected at the experimental endpoint, and thus at a later timepoint in treated mice. Nevertheless, we found that, in treated mice, the number and size of metastatic vascular emboli were significantly reduced as compared with controls (Supplementary Fig. 4k, l), indicating that trametinib can also prevent the growth of disseminated cells.

To evaluate whether sensitivity to trametinib can be predicted in CUP patients, we set up an original "trametinib response signature". To this aim, we correlated publically available data on the trametinib response of 445 cancer cell lines (CCLs) with the respective gene expression profiles[42,43] (Supplementary Fig. 4m–o). The signature ability to predict the trametinib response was first validated in an independent dataset of 634 CCLs[44] (Supplementary Fig. 4o), and further confirmed in a panel including the six agnospheres and the *BRAF*-mutated melanosphere mMS321. Here, the signature correctly predicted sensitivity in mMS321, AS901, AS906, AS43, and AS67, and resistance in AS914 and N-AS47 (Fig. 5h), accordingly to in vitro responses (Fig. 5b and Supplementary 4h). The signature was then applied to the transcriptome obtained from paraffin-embedded tissues of a retrospective cohort of patients, encompassing (i) 15 CUPs, including those matched with four agnospheres (Supplementary Tables 1 and 2); (ii) three metastatic *BRAF*-mutated melanomas (including mMEL321 that originated mMS321), chosen as positive controls for trametinib sensitivity; and (iii) ten metastases from early metastatic, non-CUP patients (i.e., enrolled as potential CUPs, but whose tissue of origin was then identified (Supplementary Table 1). The signature predicted trametinib sensitivity in all *BRAF*-mutated melanomas, as expected, and in 11/15 CUPs (73%), while it predicted sensitivity only in 3/10 early metastatic tumors of known origin. Clear resistance was predicted in the neuroendocrine CUP AGN47 (the origin of N-AS47), and in an early metastatic merkeloma (mMERK44), another tumor of neuroectodermal origin. All other cases (3/15 CUPs and 6/10 early metastatic tumors) were below the significance threshold for being defined either as sensitive

or resistant (Fig. 5i). From these data, we can conclude that (i) agnospheres and the corresponding original tumors were predicted consistently, attesting that the signature retains its ability to reliably classify suboptimal samples, such as paraffin-embedded tissues; (ii) the majority of CUP tissues, but only a minor fraction of early metastatic tumors of known origin, associated with trametinib sensitivity independently of the presence of *BRAF* or *RAS* mutations, thus suggesting that constitutive and addictive activation of the MEK signaling pathway can be sustained by alternative mechanisms, including growth factor receptor autocrine loops.

Finally, we also analyzed different synchronous metastases collected at warm autopsy from patient AGN43 (the origin of AS43)[11], showing that all share the trametinib sensitivity signature (Fig. 5j). This finding adds to the previously shown genetic homogeneity of these metastases[11] and suggests that, at least in some cases, the cell-autonomous behavior of CUP stem-like cells may translate in remarkable biological homogeneity and drug sensitivity across metastatic sites, with favorable implications for CUP therapy.

## Discussion

Isolation of stem-like cells from CUP patients provided, to our knowledge, the first CUP experimental model, suitable to identify genetic and molecular determinants that explain the disease hypermetastatic phenotype, featuring early widespread dissemination, ability to quickly colonize multiple organs by adapting to different microenvironments, and cell differentiation block.

Irrespective of their different genetic alterations, agnospheres display common properties, among which the first and most striking is proliferative and self-renewal autonomy: agnospheres retain their ability to long-term propagate without differentiating even in the absence of any exogenous growth factor, a property seldom displayed by stem-like cells derived from other metstatic tumors. We could associate this property with the cell-autonomous expression of self-renewal and reprogramming TFs, and the constitutive enrichment of a transcriptomic signature distinctive of ES cells[22]. In contrast, in ES cells, this genetic program is driven by extracellular signals, whose withdrawal causes ES differentiation[20]. Within this signature, genes controlled by Polycomb repressors[45] and by the MYC family[46] were strongly modulated, consistently with widespread expression of their transcriptional drivers (BMI1 and EZH2, and MYC, respectively). Interestingly, this ES signature was enriched not

only in agnospheres but also in original CUP tissues, indicating that the entire tumor is mostly formed by cells with stem features, and that, vice versa, agnospheres well represent the overall CUP cell population, thus behaving like "tumoroids"[29]. This also indicates that stem culture conditions used to select agnospheres do not significantly modify the genetic programs expressed by the original CUP cells.

The c-MYC proto-oncogene is well known for its role at the crossroad between stem cell regulation and oncogenesis. Famous for being included in the original Yamanaka cocktail that reprograms differentiated fibroblasts to pluripotency[47], c-MYC has been recognized as a key factor to recruit quiescent stem cells into proliferation, by providing not only direct stimulation of the cell cycle, but also activation of metabolic genes that support the bioenergetics needs of proliferating cells[48]. Overexpression of the c-MYC proto-oncogene contributes to tumor onset and/or progression, as indicated by findings in human tumors (that overexpress MYC in ~30% of cases) and mouse models[38,49]. Experimental c-MYC overexpression can confer addiction, so that MYC knock-down causes regression of established tumors, attesting its relevance for cell proliferation and survival[50].

In agnospheres and original CUP tumors, high and constitutive expression of MYC family genes (c-MYC and/or N-MYC) occurs in the absence of MYC genetic alterations. c-MYC is expressed in the agnospheres derived from the most aggressive cases, diagnosed as "poorly differentiated carcinomas of unknown origin". N-MYC alone is highly expressed in the neuroendocrine CUP, consistently with its specific role in tissues of neuroectodermal origin[51].

A second feature shared by all agnospheres, which, again, can be correlated with independency from exogenous signals, is the ability to widely metastasize after subcutaneous transplantation: indeed, agnospheres quickly disseminate, home, survive, and thrive in multiple tissue contexts, where they consistently reproduce the histology of the original tumors, including expression of markers specific to each different patient, together with lack of terminal differentiation. This adaptability is likely conferred by the ability to sustain proliferative and pseudo-differentiative programs in a niche-independent way. Importantly, our data indicate that such prerogative is passed on from patients to the experimental CUP model through agnospheres, likely as result of (epi)genetic mechanisms rendering the proliferative pathway constitutive, either through growth factors autocrine loops or altered signal transduction.

In spite of the above common traits, the agnosphere experimental behavior diverged according to the different clinical courses of the respective original patients: not surprisingly, the frequency of TICs was unusually high and growth of established metastases was rapid in agnospheres isolated from the most aggressive cases, while the same features were decidedly blunted in the N-AS47 from the long-survivor neuroendocrine CUP.

In looking for CUP liabilities, we reasoned that the variegated panel of proliferative signals detected in agnospheres should converge on constitutive activation of the MAPK pathway, known to be a major inducer of MYC expression and activity[38,39]. Interestingly, hyperactivation of the MAPK pathway has been recognized as a distinctive feature of metastatic tumors in general[52] and CUP in particular[53], together with MYC overexpression[54]. We could demonstrate that 4/6 agnospheres were addicted to the MAPK pathway and highly sensitive to the specific MEK1/2 inhibitors trametinib or selumetinib[37]. Sensitivity correlated with complete downregulation of c-MYC expression, proliferation arrest and cell death in vitro, induction of massive necrosis in experimental tumors growing at the agnosphere injection site, and reduced growth at metastatic sites. Interestingly, in many cancers displaying MAPK hyperactivation, including

those harboring RAS mutations, the administration of MEK1/2 inhibitors is poorly effective, as it interrupts negative feedbacks reverberating from the MAPK pathway to tyrosine kinase receptors. These feedbacks may include signals rapidly leading to receptor downregulation, as well as long-term kinome transcriptional reprogramming[37]. Evidence indicates that these mechanisms are likely and consistently disrupted in responsive agnospheres, as trametinib did not induce a rebound effect of tyrosine kinase receptor reactivation, but it rather induced receptor downregulation. Conversely, in non-responsive agnospheres, MEK inhibitors either failed to block cell proliferation, or paradoxically induced it. However, this is not surprising as, for example, in normal ES cells, MEK inhibitors are used to promote rather than to halt proliferation[20]. The mechanisms of resistance to MEK inhibitors remains to be determined, but they likely involve alternative signaling pathways that contribute to maintain high levels of MYC gene expression. An attractive candidate is the Wnt pathway, a main regulator of stem cell self-renewal, known to upregulate MYC[55].

To assess whether sensitivity to trametinib, a clinically approved drug, could be predicted in CUP cases, and provide a tool to stratify patients for trials, we elaborated an original "trametinib response signature". This signature correctly anticipated the experimentally assessed response to trametinib in agnospheres, and was retrieved also in the matched patients' tissues. Thus validated, the signature predicted the response in a retrospective cohort of CUP cases. Despite the absence of BRAF or RAS family mutations, usually associated with trametinib sensitivity[37], the majority of CUPs were classified as responders. Interestingly, CUP sensitivity predicted by the trametinib signature approximates that of BRAF-mutated melanomas, while a panel of aggressive metastases from carcinomas of known origin did not display a responder signature. Overall, data presented in this study indicate that constitutive activation of the MAPK pathway, leading to MYC sustained expression, and the ensuing stem and proliferative transcriptional programs, can be a CUP prominent, widespread and distinctive pathogenetic mechanism offering opportunities for therapeutic intervention.

Beyond CUP investigation, the integrated experimental platform presented in this study can have far-reaching applications, as it is endowed by unique prerogatives, compared to the metastatic models available so far[34]. First, while such models are mostly based on genetically engineered mice, or the use of conventional cells lines, either artificially manipulated to become metastatic[34], or injected in the hearth left ventricle to get systemic multi-organ spread[56], human agnospheres are innately endowed with comprehensive metastatic programs, faithfully inherited from the original tissue and passed on to the mouse model. Moreover, with their properties, agnospheres can help to overcome operational limitations recognized to current metastatic models, such as being (i) slow and inefficient, because generated late in progression by rare cell subpopulations; and (ii) based on cooperation between cell-intrinsic and environmental signals, that hamper successful colonization. The model here described thus represents a next-generation tool for functional validation of metastatic determinants, and mechanistically supported therapeutic interventions in a broad spectrum of aggressive tumors.

## Methods

**Human subjects**. All patients were recruited at the Candiolo Cancer Institute, FPO-IRCCS, according to ethical requirements and protocols approved by the institutional Review Board on human experimentation. Informed consent was obtained from all patients. All patient samples were de-identified before processing. Suspected CUP patients were enrolled in an approved prospective observational trial (Study Protocol N. 010-IRCC-10IIS-15, and following updates; last update: v2.0-16.10.2018), where the diagnosis was attained through an "ad excludendum"

diagnostic protocol in accordance with ESMO guidelines[2]. After diagnosis, patients received the best standard of care, and provided archival and viable tissue specimens, and blood samples.

**Animal models**. All animal procedures were performed according to ethical regulations and protocols approved by the Italian Ministry of Health. NOD.CB17-Prkdc[scid]/NcrCr mice (NOD/SCID), (RRID:IMSR_CRL:394, Charles River Laboratories), 5- to 6-week-old male were used for all in vivo studies. Mice were housed at a maximum of six per cage with a 14-h light/10-h dark cycle, in a conventional animal facility with an ambient temperature and humidity of 20–26 °C and 40%–60%, respectively, with food and water ad libitum. Mice were monitored at a minimum of twice weekly for general performance status and euthanized when volume of xenografts reached 1600 mm$^3$, or they displayed signs of distress, or weight loss ≥ 20%.

**Generation and culture of agnospheres and tumorspheres**. Tumor tissue specimens derived either from CUP patients (AGN906 and AGN47), or PDX (AGN901, AGN43, AGN67, and AGN914), or a melanoma patient (mMEL321) were minced and digested with collagenase I (Gibco) at 2 mg/ml in culture medium for 40 min at 37 °C. After filtration and red cell lysis, single-cell suspensions were resuspended in culture medium, composed by: DMEM:F12 (Sigma), N2 supplement (Life Technologies-GIBCO), BSA 0.5% (Sigma), heparin 4 µg/ml (Sigma), 2 mM glutamine (Sigma), penicillin–streptomycin (EuroClone), and seeded in ultra-low-attachment flasks (Corning-Sigma). For AS914 and AS67, the culture medium was supplemented with a chemically defined Lipid Concentrate (Gibco). Culture medium was replaced the next day and used throughout propagation. Spheroids (agnospheres) appeared in culture few days after seeding, and stabilization occurred on average after 3–4 months. For dissociation, trypsin was required for all agnospheres except for AS906 and AS914. mCRC729 and mCRC0155, previously derived from colorectal cancer liver metastases[17], were kept in the same culture medium as above, supplemented with Lipid Concentrate (Gibco). Agnospheres and tumorspheres will be available upon institutional material transfer agreement approval.

**Generation of patient-derived xenografts**. Human CUP tumor specimens, derived either from biopsy or surgery, were subcutaneously transplanted in NOD/SCID mice (PDX). Tumors were explanted when reached a maximum of 1600 mm$^3$, and collected for further propagation and agnosphere derivation.

**Histopathology**. Immunohistochemical staining of formalin-fixed paraffin-embedded (FFPE) tumor tissue sections derived from patients and animal models was performed using Ventana Benchmark ultra System (Roche), or Bond Max (Leica Biosystems) or Autostainer Link 48 (Agilent), according to the antibody used (see reagents), and revealed with Liquid Diaminobenzidine (DAB) + Substrate Chromogen System (K3468, Dako), following standardized manufacturers' instructions. For quantification of experimental tumor areas, stainings of representative middle tumor sections collected at the experimental endpoint were acquired with Nikon Eclipse Ti2 using the NIS Elements Imaging software. The whole area was scanned at 4× magnification and analyzed with ImageJ (RRID: SCR_003070). Percentage of hematoxylin-positive area or Cyclin D1-positive area was normalized vs. total area for each sample. Necrotic areas were calculated as total area minus vital area. Statistical significance was assessed by two-way ANOVA (Sidak's multicomparison test).

**gDNA extraction and CNV analysis**. Genomic DNA was isolated from agnospheres using Relia Prep TM gDNA Tissue Miniprep System (Promega), according to manufacturer's instructions. DNA was quantified using a Nanodrop ND1000 spectrophotometer (Thermo Fisher Scientific). As normal control, gDNA of PBMCs derived from a pool of five healthy individuals was used. *PTEN*, *TP53*, *c-MYC*, and N-*MYC* copy-number variations were calculated after qPCR with the $2 - \Delta\Delta Ct$ method using *GREB1* as normalizer.

**Mutational screening of FFPE tissue specimens**. In tumor samples not analyzed by WES, assessment of mutational status of *KRAS*, *HRAS*, *NRAS*, and *BRAF* hotspot mutations was performed through OncoCarta™ Panel v1.0 (Agena Bioscience) on gDNA extracted from FFPE tissues, according to manufacturer's instructions.

**Chromosome G banding**. Chromosome analysis by G banding was performed on agnospheres. In order to increase the number of metaphases, cells were synchronized with Synchroset (Euroclone S.p.A.) according to manufacturer's instructions and blocked with colcemid (10 µl/ml) for 1 h. Chromosome harvesting was carried out according to standard procedures. Briefly, cells were incubated in 0.075 M KCl hypotonic solution at 37 °C for 10 min, fixed in methanol–glacial acetic acid (3:1), dropped onto glass slides and dried using specific conditions for optimal chromosome spreading. G banding was performed by incubating slides in 2× SSC at 68 °C for 2 min and eventually stained with Wright's solution for 2 min (ref. [57]). Metaphase images were captured using an Olympus BX61 microscope (Olympus Corporation)

and analyzed by CytoVision software (Leica Biosystems). An average banding resolution of 300 bands was achieved. Aberrations were described according to the International System for Human Cytogenetic Nomenclature, 2016 (ref. [58]).

**Chromosome M-FISH**. Chromosome analysis by multicolor-fluorescence in situ hybridization (M-FISH) was carried out on agnospheres by using a commercial mixture containing 24 chromosome-specific painting probes labeled with different fluorochromes (24×Cyte kit MetaSystems). Probes and metaphase chromosomes denaturation and hybridization were performed according to manufacturer's instructions. Briefly, the slides were incubated at 70 °C in saline solution (2× SSC), denatured in NaOH, dehydrated in an ethanol series, air-dried, then covered with 10 µl of the denatured probe cocktail and hybridized for 24 h at 37 °C. Subsequently, the slides were washed with post-hybridization buffers and counterstained with 10 µl of DAPI/antifade. The Metafer System and the Metasytems ISIS software (Carl Zeiss) were used for signal detection and metaphase analysis. At least nine metaphases along different culture passages exhibiting the same derivative chromosomes were studied for each cell line.

**RNA-seq and qRT-PCR**. Growing agnospheres and tumorspheres were harvested and RNA was extracted using RNeasy Micro Kit (Qiagen), following manufacturer' instructions. For qRT-PCR mRNA was converted into first-strand cDNA using superscript II Reverse Transcriptase (Invitrogen), according to manufacturer's instructions. Amplification was performed with ABI PRISM 7900 HT (Applied Biosystem) using Taqman Probes (Applied Biosystem). Ct values were normalized versus HPRT1 and represented in the heatmap as 40 − Ct. Results are representatives of at least two independent experiments.

For Quant-seq 3′ mRNA sequencing, total RNA from agnospheres and tumorspheres or from FFPE patient tissues was purified using RNeasy Micro Kit (Qiagen) or Maxwell® RSC RNA FFPE Kit (Promega), respectively. Total RNA was quantified using the Qubit 2.0 fluorimetric Assay (Thermo Fisher Scientific). Libraries were prepared from 100 ng of total RNA using the QuantSeq 3′ mRNA-Seq Library Prep Kit FWD for Illumina (Lexogen GmbH). Quality of libraries was assessed by using screen tape High-sensitivity DNA D1000 (Agilent Technologies). Libraries were sequenced on a NovaSeq 6000 sequencing system using an S1, 100 cycles flow cell (Illumina Inc.).

**Alignment and normalization of QuantSeq RNA data**. Illumina novaSeq base call (BCL) files were converted in fastq file through bcl2fastq (version v2.20.0.422, Illumina Inc.). Sequence reads were trimmed using bbduk software (bbmap suite 37.31, Joint Genome Institute) to remove adapter sequences, poly-A tails, and low-quality end bases (regions with average quality <6). Alignment was performed with STAR 2.6.0a (RRID:SCR_015899)[59] on hg38 reference assembly obtained from cellRanger website (Single-Cell Gene Expression, 10× Genomics; Ensembl Assembly 93). The expression levels of genes were determined by htseq-count 0.9.1 by using cellRanger pre-build genes annotations (Single-Cell Gene Expression, 10× Genomics; Ensembl Assembly 93). Genes with an average number of CPM (counts per million) <5 and Perc of duplicated reads >20% were filtered out. Data normalization were performed using edgeR (RRID:SCR_012802)[60].

**Analysis of embryonic stem cell signature**. Transcriptional profiles of agnospheres and tumorspheres were first averaged across replicates, and then genes with an average CPM < 1 across all samples removed. Then averaged transcriptional profiles of spheres and transcriptional profiles of CUP tissues were converted in z-score and the gene set enrichment analysis (GSEA)[61] was performed using as input the ES signature as described by Ben-Porath et al.[22]. The expression profiles and clinical information of 942 breast cancer patients described by Ben-Porath et al.[22] were downloaded from barc.wi.mit.edu/benporath. Patients' transcriptional profiles were first converted in z-scores and GSEA performed as above. Then, median enrichment score of each gene signature was plotted grouping patients according to their breast tumor grade. Finally, gene expression profiles of human ES cell lines H7, HUES1, HUES8, and HUES9 were retrieved from ref. [62] with accession number GSE102311, and GSEA was performed after transcriptional profiles were converted in z-scores. GSEA and associated statistics were computed using the fgsea package in R statistical environment version 3.6.

**Generation and validation of the trametinib response signature**. The basal expression profile of ~1000 CCLs was obtained using RNA-seq from Cancer Cell Line Encyclopedia[24,43] (https://portals.broadinstitute.org/ccle). Cell lines derived from liquid tumors were discarded, and only cell lines derived from solid tumors were used for the following analysis. The raw counts of each gene across the cancer cell lines were normalized using edgeR[63]. Data on cell line response to trametinib were previously generated by Rees et al.[42] and expressed in terms of area under the curve (AUC), which reflects the in vitro response to trametinib of each cancer cell line[24] over 72 h. In particular, lower values of AUC are associated with a higher sensitivity to trametinib and vice versa. A total of 445 CCLs, for which both trametinib response and expression data were available, were used to identify marker genes associated to trametinib resistance, as described below. The gene signature to predict trametinib response from transcriptional data was identified as depicted in Supplementary Fig. 4m–o. First, we computed Pearson correlation

coefficient (PCC) between the expression of each gene and trametinib potency measured by mean of AUC across the 445 distinct CCLs selected, as described above. We then considered the top 1000 genes with highest PCC as putative marker genes of trametinib resistance, because higher expression of these genes is associated with lower potency of trametinib across the panel of the 445 used CCLs. Finally, a machine-learning approach based on recursive feature elimination (RFE) and support vector machines (SVMs) was used to identify the 500 (out of 1000) genes whose expression values were best at discriminating resistant from sensitive trametinib CCLs[64,65]. In particular, we used linear SVMs that were trained and tested using the kernlab package in the R statistical environment version 3.6 (ref. [66]). For RFE, trametinib sensitive CCLs were defined as those which AUC values were in the 5% quartile of trametinib AUC distribution across the 445 CCLs used in this study. On the other hand, trametinib resistant CCLs were defined as those whose AUC values were in the 95% quartile. We used GSEA to predict the trametinib response by checking whether marker genes of trametinib resistance identified as above were either downregulated or upregulated across a given transcriptional profile (Supplementary Fig. 4m). Hence, in each transcriptional profile, genes are first sorted form the most to the least expressed and then GSEA is performed using as input the identified marker genes of trametinib resistance. A positive enrichment score means that genes associated with trametinib resistance are highly expressed in the transcriptional profile, thus predicting resistance of these cells to trametinib treatment. Conversely, a negative enrichment score indicates a low expression of marker genes of trametinib resistance, and thus it predicts sensitivity to trametinib treatment (Supplementary Fig. 4n). Validation of the method was performed by predicting the trametinib response in an independent dataset of 634 CCLs from Garnett et al.[44] (Genomics of Drug Sensitivity in Cancer: https://www.cancerrxgene.org/), for which basal gene expression from microarray and trametinib response in term of IC50 were available. We then computed the percentage of correctly predicted sensitive and resistant CCLs (Supplementary Fig. 4n). Specifically, CCLs sensitive to trametinib were defined as those with an IC50 < 1 μM, while all the other were considered as resistant. We obtained an average classification accuracy of ~76% across the two classes (sensitive and resistant) of CCLs. In all the analyses described above GSEA and associated statistics were computed using the fgsea package in R statistical environment version 3.6.

**Analysis of trametinib response signature**. Agnospheres, tumorspheres, or patient tissues transcriptional profiles were first averaged across replicates when present, then genes with an average CPM < 1 across all samples filtered out. Before applying GSEA, averaged transcriptional profiles of each sample were converted in z-score. GSEA[61] was performed using as input identified marker genes of trametinib resistance with the fgsea package in R statistical environment version 3.6.

**Immunofluorescence and cell immunohistochemistry**. Samples undergoing immunofluorescence were either FFPE tumor specimens or growing agnospheres or colosphere mCRC729. The latter were harvested, fixed 10 min with PFA 4% at 4 °C, washed in PBS, suspended in bio-agar for cyto-inclusion (Bio-Optica) at 42 °C, and processed for inclusion in paraffin. All staining were performed as previously described[67]. Images were acquired using a LEICA SPEII confocal microscope, equipped with a 40× oil objective and a 1.5× zoom for a final magnification of 600×. Optical single sections were acquired with a scanning mode format of 1024 × 1024 pixels. Fluorochrome unmixing was performed by acquisition of automated-sequential collection of multichannel images, to reduce spectral crosstalk between channels. For immunohistochemical staining, an additional peroxidase blocking was performed in $H_2O_2$ 3% methanol 50% incubated 20 min in the dark. For primary and secondary antibody concentrations, see reagents. Secondary antibodies were HRP-conjugated (Dako), and DAB substrate chromogen kit (Dako) was used for detection. Nuclei were counterstained with hematoxylin. Images were acquired through LASV4.2 software and are representative of at least three independent immunostainings.

**Western blot analysis**. Total protein were extracted using RIPA buffer supplemented with a protease inhibitor cocktail (Roche Life Science), $NaVO_3$ 1 mM, and NaF 1 mM, subjected to sonication, quantified using BCA methods (Pierce), and 20 μg were separated on SDS–polyacrylamide gradient gel 4–12% or 4–20% (Invitrogen) and blotted onto nitrocellulose membrane. After blocking, primary antibodies were incubated at the indicated concentrations (see reagents). After incubation with HRP-conjugated secondary antibodies (Jackson Lab), enhanced chemiluminescence (Biorad) was used for detection according to manufacturer's instructions and images were acquired with the ChemiDoc Touch™ Imaging System (Biorad) through Image Lab software. β-Actin or β2-tubulin were used as protein loading control as indicated. The results shown are representative of at least three independent experiments.

**In vitro limiting dilution assay**. Agnospheres were dissociated and seeded at limiting dilution concentration (1–100 cells/100 μl) in ultra-low-attachment 96-well microtiters (Corning). "Positive tests" were defined as wells with primary spheres with a diameter ≥ 100 μm. Stem cell frequency was calculated using the ELDA software[68] (http://bioinf.wehi.edu.au/software/elda/). Means and 95%

confidence intervals (CI) are shown (n ≥ 3 independent experiments). Primary spheres were harvested, dissociated, and seeded at the same dilutions for a second and a third passage to assess long-term propagation.

**Cell viability**. Agnospheres were dissociated and seeded in culture medium in 96-microtiter wells at the concentration of 200 cells/100 μl in the case of AS901, AS906, AS43, and AS67, 500 cells/100 μl in the case of AS914 or 1000 cells/100 μl in the case of N-AS47. Trametinib (GSK1120212; Selleckchem) or vehicle (DMSO) were added immediately after seeding and every 5 days at the indicated concentrations. ATP consumption was measured 4 or 10 days after cell seeding with Cell Titer Glo (Promega) according to manufacturer' instructions, using GloMax 96 Microplate Luminometer and GloMax®-96 software (Promega). In each experiment, the average of relative luminesce values (n ≥ 6 technical replicates) was normalized vs. untreated control and fold changes were reported (n ≥ 3 mean ± SEM, ANOVA, Bonferroni multicomparison test).

**In vivo agnosphere transplantation**. To assess the tumorigenic potential, agnospheres were dissociated as single-cell suspensions and counted with trypan blue to exclude dead cells. A total of $10^5$ cells were resuspended in 50 μl of culture medium mixed 1:1 with matrigel—growth factor reduced (BD Bioscience) and injected subcutaneously in the flank of 5–6-week-old male NOD.CB17-Prkdcscid/J mice (referred to as "spheropatients"), obtaining 100% engraftment efficiency. During the procedure anesthesia with 2.5% isofluorane in 100% oxygen at a flow rate of 1 l/min was delivered to mice. Tumors grown at the injection site (IS-tumors) were explanted and FFPE to undergo histopathological evaluation and comparison with original tumors, as detailed above for patients.

**Tumor-initiating cell frequency (in vivo limiting dilution assay)**. For in vivo LDA, single-cell suspensions of $10^5$, $10^4$, $10^3$, $10^2$, or 10 cells were subcutaneously injected into immunocompromised mice (P0) as above. Tumor intake (formation of IS-tumors) was monitored up to 7 months after transplantation or until spontaneous death, and the in vivo TIC frequency was calculated with the ELDA software as above. Mice dead before the experimental endpoint that did not generate a tumor were excluded from the counts. From P0 tumors agnospheres were re-derived in culture by enzymatic digestion with collagenase as above, and, after 1 week of recovery in culture, they were dissociated and transplanted for a second passage in mice (P1) at the same cell dilutions as above[69]. In some experiments, the procedure was further repeated for a third passage (P2).

**Assessment of spontaneous metastases**. Agnospheres and colosphere mCRC729 were engineered to express luciferase. Briefly, single-cell suspensions were seeded in ultra-low-attachment six-well microplates at $5 \times 10^5$ cells/well concentration in 1.5 ml of culture medium. Lentiviral particles carrying pCMV-Luciferase or pCMV-Luciferase-IRES-GFP transfer vectors were added at 5 MOI. After 6 h, 1 ml medium was added, and 24 h later the medium was replaced. Transduction efficiency was assessed at 72 h by fluorescence microscopy showing 98% in all agnospheres. Transduced agnospheres were dissociated to single-cell suspensions, and $5 \times 10^4$ cells were injected as detailed above in the flank of NOD/SCID mice. Within a month after transplantation, tumors reached ~300 mm³ and were surgically removed under anesthesia (as above) to prevent tumor overgrowth during subsequent metastasis monitoring. Tumor volume was measured with a caliper and calculated using the formula: $(d)^2 \times (D)/2$, where d and D are the minor and the major tumor axis, respectively. Analgesics were delivered to the animals according to institutional guidelines. Bioluminescence in vivo imaging was acquired with IVIS® Lumina and IVIS imaging software (Caliper Life Sciences) starting from the day after surgery. Luciferin (D-Luciferin potassium salt, Caliper Life Sciences) dissolved in PBS (150 mg/kg) was administered to mice by subcutaneous injection. Anesthesia was delivered in the induction chamber with 2.5% isoflurane in 100% oxygen at a flow rate of 1 l/min and maintained during IVIS imaging with a 1.5% isoflurane mixture as above at 0.5 l/min. To mask residual signals from the subcutaneous tumor, a black tape was applied during imaging. Luminescent signals were monitored weekly until mice reached the clinical endpoint. For ex vivo imaging, luciferin was administered as above, mice were euthanized by carbon dioxide inhalation and all organs were immediately explanted and analyzed with IVIS. Organs were FFPE to undergo histopathological evaluation and comparison of original tumors as above. The frequency of metastatic sites was assessed by ex vivo organ luminescence and by immunohistochemistry of CUP-specific markers in organ sections. Such frequency could be underestimated for missed identification of micrometastases in some organs. To assess early dissemination, mice transplanted with AS43 were euthanized 10 days after subcutaneous injection, and explanted organs were FFPE and analyzed as above.

**In vivo trametinib treatment**. Unlabeled agnospheres were inoculated subcutis in immunocompromised mice. When tumors reached ~150 mm³, mice were randomized in two groups (control and trametinib-treated) using LAS software[70]. For treatment, trametinib was dissolved in hydroxi-methylcellulose 0.5%-Tween 80 and administered by oral gavage at 1 mg/Kg/die (therapeutic range in preclinical models: 0.3–3 mg/kg/die)[41]. Tumor volume was measured and calculated as above,

and expressed as fold changes vs. day 0 for each mouse. For each experiment, $n \geq 6$ mice/group were used. Statistical significance was determined by two-way ANOVA. The survival curve was generated considering a tumor volume of 1600 mm$^3$ as experimental endpoint, and statistical significance was calculated using log-rank (Mantel–Cox test).

**Statistical analysis**. Results were expressed as mean ± standard error of the mean (SEM) of at least three independent experiments. Statistical comparisons were performed using Prism v8.0 software (RRID:SCR_005375; GraphPad). Chi-squared test was used for limiting dilution experiments performed by ELDA software. ANOVA (Bonferroni's multiple comparison test correction) was used for assessment of in vitro trametinib and selumetinib effect. Two-way ANOVA was used for tumor growth curves, Mantel–Cox test for survival curves and two-way ANOVA (Sidak's for multiple comparisons) for evaluation of vital/necrotic tumor areas. A $p$ value <0.05 or a false discovery rate < 10% was considered significant. For RNA-seq analysis the fgsea package in R statistical environment version 3.6 was used.

**Reporting summary**. Further information on research design is available in the Nature Research Reporting Summary linked to this article.

## Data availability

Complete datasets related to WES analysis of somatic mutations, referring to Supplementary Table 3, were deposited in the European Genome-phenome Archive (EGA), under the accession code EGAS00001004868. Raw data relative to 3′UTR-Seq, referring to Figs. 2d and 5g–I and Supplementary Fig. 1d, e were uploaded in the Gene Expression Omnibus (GEO) repository, under the accession code GSE167473. Datasets from public databases are available at these web links: Cancer Cell Line Encyclopedia: https://portals.broadinstitute.org/ccle; Genomics of Drug Sensitivity in Cancer: https://www.cancerrxgene.org/. Source data are provided with this paper.

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

## Acknowledgements

We thank P. Luraghi for mCRC729 and mCRC0155 generation, S. Kolling, F. Ventre, and A. Polidori for help with patients' data, the TIGEM Bioinformatics Core for help with bioinformatic analysis, I. Sarotto, M. Milan, P. Ferrero, R. Albano, S. Giove, and R. Carollo for technical help, the TIGEM NGS facility for sequencing, C. Marchiò for organizational help, A. Balsamo, A. Cignetto, and D. Gramaglia for assistance. This work was supported by AIRC—Italian Association for Cancer Research ("Special Program Molecular Clinical Oncology 5 × 1000", N. 9970 and N. 21052, and Investigator Grant N. 19933 to C.B.); FPRC 5 × 1000 "2014" and RC "2019", Ministero della Salute; F.V. and E. Candiello were recipients of "Fondazione U. Veronesi" postdoctoral fellowships. G.G. was supported by the STAR (Sostegno Territoriale alle Attività di Ricerca) grant of University of Naples Federico II and My First AIRC Grant 23162.

## Author contributions

F.V. and C.B. conceived the project, planned experiments, interpreted data, and wrote the manuscript with inputs from all the authors; F.V. was involved in performing all the experiments; G.G. performed bioinformatic analysis; A.P., E. Cascardi, A.S., and R.S. performed histopathological diagnosis in patients and models; A.D.A. helped with in vivo, and E. Candiello and M.F. with in vitro experiments; M.P. and L.C. analyzed cytogenetics; S.B. contributed to genetic analysis; E.G. and F.M. managed patients; A.B. supervised bioinformatic analysis; and P.M.C. and C.B. supervised the project.

## Competing interests

A.B. is a cofounder of CASMA Therapeutics Inc. All other authors declare that they have no competing interests.
