## [Peer Review File · Nature Communications]

Reviewers' Comments:

Reviewer #1:

Remarks to the Author:

In their manuscript Verginelli and coworkers present evidence suggesting that cancers of unknown primary (CUP) represent a distinct pathological entity originating from the malignant transformation of stem-like cells. For their experiments they have generated patient-derived xenografts (PDX) and spheroid cultures (agnospheres) from tumor tissue of six and five patients with CUP syndrome, respectively. Using these tools they describe that agnospheres behave as CUP-initiating, stem-like cells with cell-autonomous, early dissemination and widespread metastasis. This phenotype resulted from constitutive activation of MAP kinases and MYC, leading to sensitivity towards MEK inhibitors. They conclude that MEK inhibition might constitute a novel treatment paradigm in CUP syndrome as a transcriptomic signature predicting response towards MEK inhibitors was found in 75% of CUP patients. Although the manuscript is interesting and concerns a clearly understudied cancer type, it suffers from several shortcomings that need to be taken care of before publication:

Major comments:

- (1) A consistent control group, representing at least some of the major cancer types, that is stringently used throughout all experiments shown in the manuscript needs to be established. In Figure 2, only CRC729 (colorectal cancer) and MS321 (melanoma) are shown as controls in panel A, no control is displayed in panel B, only CRC729 is shown in panel C, CRC729, CRC0155 (typo in panel D) and MS321 are used in panels D - F, and no control is shown again in panel G. In contrast, the figure legend of Figure 2 lists explanations for abbreviation of glioblastoma, colorectal cancer, pancreatic cancer, melanoma and leukemia.
- (2) Along the same lines, for the experiments shown in Figures 3 - 5, with the exception of single controls in Figures 4E (CRC729) and 5A (MS321), no controls are given at all. Instead, organoids derived from aggressive metastases of known origin need to be used as comparators here.
- (3) Figure 5: The effect of trametinib on metastases should to be evaluated in addition to its effect on the injection site tumors.
- (4) In the abstract of their manuscript the authors state that sensitivity to MEK inhibitors "is associated with a transcriptomic signature predicting that 75% of CUP patients could be eligible to MEK inhibition". This conclusion derives from the transcriptomic analysis of paraffin-embedded primary tumor tissue from only 12 CUP cases (including those matched to the four agnospheres used in the manuscript) and without any patient having been treated with trametinib (for example on a compassionate use basis), and seems therefore clearly overstated.
- (5) Page 8: The authors describe that EMT transcription factor expression was relatively weak in adeno-CUPs but high in CUPs derived from carcinomas with sarcomatoid features. This finding obviously argues against a common CUP pathophysiology and phenotype and against a major role of the EMT for metastasis in CUP as the vast majority of CUPs are adenocarcinomas without displaying sarcomatoid features.

Minor comments:

- (1) The manuscript needs language editing.
- (2) On page 6 it is stated that some of the agnospheres harbored mutations "some of which are among the ten most frequently mutated genes in CUPs, being enriched in CUPs compared with metastases from cancers of known origin". This suggests that CUP-specific mutational spectra exist, which is not the case.

Reviewer #2:

Remarks to the Author:

This study addresses an important issue in the cancer field, which is modeling, characterizing and finding therapeutic targets for Cancer of Unknown Primary (CUP). This is one of the few groups in

the world that have addressed this intractable but actually quite prevalent type of cancer diagnosis so it is a meritorious effort that may have significant clinical impact.

Verginelli and colleagues isolated human CUP cells from patient samples and PDX models and successfully generated spheres, named 'agnospheres'. These agnospheres harbored the same genetic alterations found in patients' tumors, as copy number variations in TP53 and PTEN. Further characterization by different approaches (RT-qPCR, Western blot, RNA-seq) revealed expression of transcription factors regulating stem cell self-renewal and maintenance. Enrichment of an embryonic stem cell expression signature in the RNA-seq supports the finding that agnospheres derived from CUP samples are composed of stem-like cells.

The most interesting findings of this work come from the transplantation of agnospheres in mice. As few as 10 cells were able to generate subcutaneous tumors, indicating a potent tumor-initiating ability of agnosphere-derived cells. Importantly, the established tumors faithfully recapitulated early dissemination and metastatic spread to multiple organs, as seen in CUP patients. The authors further found that growth was dependent on MEK1/2 and agnospheres were sensitive to MEK1/2 inhibition via Trametinib treatment both in vitro and in vivo.

Overall, the study is innovative in the field and provides a first and powerful model to gain insight into the biology and therapeutic approaches to target CUP.

Some technical concerns and further experiments to strengthen the findings are listed below:

Major points:

1. In Figure 2, the authors provide RT-qPCR, Western blotting, Immunofluorescence and RNA-seq data to support the hypothesis that agnospheres express stem cell markers and factors. However, in panels A-C, it would be more convincing to include a positive control and a negative control (e.g. a tissue-derived or in vitro induced stem cell as positive and another cancer cell – patient-derived or cell line – as negative). The data in D and E is displayed in a confusing way. It would be useful to see how the signatures and genes compare to ESC, rather than having exclusively absolute values and scores from the RNA-seq analysis. Most importantly, the media used for agnosphere formation naturally enriches for stem cells, as there are selected to survive in this condition. Thus, detection of these markers from fresh tumors would be helpful. Alternatively, qPCR from RNA from FFPE samples or IF on the sections may be a good validation of the markers in the patient samples. Having other less aggressive type of cancer cells in the same media would provide an insightful comparison to rule out the argument that what is seen is not specific to CUP agnospheres but due to the culture conditions. For example tumor from the same tissue that did not present as CUP. Agnospheres could also be dissociated, removing the cells from stem-cell media and placing them in regular media. Do they still display/retain stem-like properties?
2. In Figure 5, the authors explore the effect of Trametinib on agnospheres in culture and in the CUP preclinical model. ATP consumption should be measured at the same timepoint for all agnospheres (some are measured at 4 days, others at 10 days). As shown in Supplementary Figure 5, AS914 agnospheres had increased proliferation in response to Trametinib, in contrast to the other patients' spheres. From the characterization and RNA-seq, the authors should attempt to find what is specific to that sample compared to the others to provide an explanation for this result. Other Mek inhibitors should be tested or genetic approaches to KD Mek and ERK should be employed to explain the differences. Mutational analysis of the Ras pathway should also be provided to rule out that as in AS47 line the lack of sensitivity is not due to a super-active Ras mutant pathway.
3. To assess Trametinib effect in vivo, they monitored the response on subcutaneous tumors. Although the authors provide multiple explanations for this choice and the results are convincing, it is necessary to look at secondary organs. CUPs do not present as a primary and the multiorgan metastasis patterns may also reveal differenced in the response to MEK1/2 inhibition that could be highly informative for potential clinical trials. Show tumor volume change until end of treatment. Further, to reproduce what seen in CUP patients, the authors should first remove the primary tumor by surgery, then starting Trametinib treatment and evaluating the effect on metastatic outgrowth. Also since in some animals dissemination occurs almost immediately after implantation

it could be tested if MEK1/2 is also needed for this process as it may also provide insight into how this pathway is wired for dissemination and growth in these tumors.

4. Although Trametinib is an interesting candidate compound from this analysis, an alternative therapeutic strategy would make the work stronger (as potential acquired resistance has not been addressed in detail and serious side effects have been shown in patients). Going back to the RNA-seq to identify novel targets specific to CUP samples would bring a more innovative and impactful side to the study. For example WBs in Figures 2 and 5 offer other alternative treatments such as inhibitors or mAbs targeting surface receptors such as EGFR, HER2 IGFR and Met. These are heterogeneously activated in the different agnospheres but not much explanation is provided as to why these are different. Also two agnospheres including a MEK1/2 inhibitor-resistant tumor seem to upregulate EZH2 and there are excellent therapeutics for this chromatin remodeler. It seems that as CUP is very diverse their mechanism of growth promotion may be diverse and limiting it to Mek1/2 inhibition may be less effective. Testing these other targets with drugs that are available may be useful

Minor points:

1. In Figure 2, please include insets of immunofluorescence images, particularly where the arrows are pointing to co-localization.
2. Figure 3 demonstrates the high tumorigenic capacity of agnospheres, with an elegant in vivo limiting dilution transplantation experiment. I suggest to include in the experiment a control of spheres generated from other cancer cells (as shown later for melanospheres and colospheres) from the same tissue of origin but not derived from CUP.
3. In Figure 5 and Supplementary Figure 5, relative ATP consumption by Cell Titer Glo kit is used to assess agnosphere viability. Agnospheres were dissociated and seeded as single cells in the experiment. Cell viability should also be measured by treatment of agnosphere cells in full, non-dissociated spheres to establish whether trametinib has the same effect.
4. A genetic proof of MEK dependency in agnospheres must be used to strengthen what was found with pharmacological Trametinib treatment. Using siRNA or CRISPR to eliminate MEK and ERK in the case of AS47 in these cells will show specificity and selectivity.
5. On pages 12/13, in the Results section, the authors state that "single disseminated cells were identified by pan-cytokeratin immunostaining, thus retaining their epithelial phenotype, suggesting that EMT, if ever occurred, rapidly disappeared". I suggest rephrasing this sentence as expression of pan-cytokeratin does not exclude that these cells are still residing in a partial EMT state.
6. Further below in the text, the authors claim that the inability to form detectable metastases from colospheres injected subcutaneously "is consistent with lack of evidence in similar models reported by the literature". Please include citations to support the statement.

Reviewer #3:

Remarks to the Author:

This manuscript characterizes Cancers of Unknown Primary (CUPs), and identifies a potential novel treatment for these aggressive cancers. The authors established agnospheres (spheroids propagated in suspension) from metastatic CUP patient samples (in vitro) and agnosphere-PDX models (in vivo) from 6 CUPs. The authors use these models to demonstrate that the agnospheres have stem-cell traits and activation of the MEK/ERK/MYC signaling pathway. Moreover, they show that the agnospheres are highly sensitive to the MEK inhibitor, trametinib, and identify a trametinib sensitivity signature. Strengths of this manuscript include the significance of studying this class of patients who have few treatment options. In addition, the manuscript includes a plethora of sequencing/bioinformatics analyses, which provide a good foundation for mechanistic studies. Moreover, the authors utilize in vitro and in vivo experiments based on patient-derived material to evaluate their hypotheses. Furthermore, most experiments are well-controlled, data are clear, the manuscript is well-written, and statistical measures are robust, in most instances. However, despite these strengths, the manuscript also contains a number of weaknesses, which

detract from the strengths. Firstly, although the authors perform Whole Exome sequencing on the agnospheres, they never link genomic alterations with pathway activation identified in subsequent in vitro and in vivo studies. Secondly, much of the data shown are merely correlative. The mechanistic data that the authors provide do not adequately or definitively link the MEK/ERK/MYC pathway to CUP proliferation/survival, stem cell features and trametinib sensitivity as important rescue experiments are lacking, and the authors do not provide corroborating evidence for the importance of this pathway in primary patient samples (uncultured; "original tissues") and in animal tumors from the in vivo studies. In addition, the in vitro experiments utilize long timepoints (48-96h for blots) and relatively high trametinib doses (50nM), which makes it difficult to assess a direct and specific effect on signaling pathway. Furthermore, the trametinib dose utilized for in vivo studies (1mg/kg), although used in some published studies, translates to 4.8mg/day in humans,¹ which is significantly above the Recommended Phase 2 Dose and thus, the dose utilized in the clinic (2mg/day), which may reduce the clinical impact of the study if this dose cannot be established in patients. Finally, although the investigators make the argument that the IS tumors that are generated mirror the growth of metastatic cells, the effect of trametinib on colonization of other organ sites (in vivo metastatic progression) and stem cell expansion (in vitro or in vivo), which the authors argue are critical characteristics of CUPs, is never assessed. Additional detail and comments are described below.

1. The data are incomplete. Some figures show data for 6 agnosphere cultures, while others show 4, 3 or 2 lines. The authors need to be consistent and show all of the data either within the figure or as Supplementary data. See additional details below.

2. Fig. 2C. AS67 and AS914 are missing from the blots. Moreover, N-AS47, the neuroendocrine tumor which displays non-metastatic features, harbors high levels of N-MYC which is contrary to the authors' hypothesis that MYC signaling drives aggressive CUP disease.

3. Fig. 5B. AS67, and AS914 are missing from the blots. Moreover, other controls such as the colon cells used in "A" are missing for comparison purposes.

4. Viability assays (CellTiter Glo), performed in Fig. 5C and other figures (e.g. S5D,E), are performed at too long of a timepoint. Viability is most often measured 72h after treatment. Here, the authors measured viability after 4-10 days. Using a prolonged treatment time complicates interpretation of the data, as media/drugs would have to be changed at least once over the course of the experiment, which can alter proliferation rates, and thus, the results. These experiments need to be repeated with shorter treatment times (72h).

5. Fig. 5D. a) AS67 and AS914 are missing from the blots. This is particularly important since trametinib treatment has opposite effects in AS914 (Fig. S5G). The authors also indicate that the paradoxical effects on AS914 are likely mediated by "rebound effect on upstream proliferative signals; pg. 15); however, mechanistic data to prove this hypothesis are never shown. b) pMEK blots are missing. This is critical given that the authors want to make the case that the MEK/ERK/MYC pathway is upregulated in these cultures, which is then targeted by trametinib. c) The blots (as well as those in Fig. S5F) were performed on lysates from cells treated for 48-96h. This timepoint is too long as it gives time for the cells to adapt and activate positive and negative feedback loops. The blots need to be performed at a short timepoint (<24h) to assess immediate drug responses. d) The authors demonstrate that trametinib treatment inhibits EGFR activation in two cell lines; however, they never identify the mechanism nor demonstrate that this downregulation (or expression of any RTK for that matter) is critical for trametinib sensitivity. The authors indicate that the mechanism of trametinib sensitivity may be tied to the finding that trametinib reduces EGFR activation, and thus, due to lack of ERK-mediated negative feedback regulation of EGFR. However, trametinib reduces pEGFR in both AS901 and AS43 cells (Fig. 5D) even though AS901 is insensitive to EGFR inhibition, whereas AS43 is dramatically sensitive to lapatinib/afatinib (Fig. S5D). Again, mechanistic experiments aimed at linking EGFR to MEK/ERK activation, trametinib sensitivity, and proliferation/survival are not complete.

6. Fig. 5E,F. a) Tumor growth curves are plotted as “fold change”. Since this removes critical information, the data instead should be shown as raw numbers (Tumor volume—mm³). b) In order to determine whether tumor growth is different between trametinib and vehicle-treatment mice, the growth rates should be analyzed with statistical methods such as linear mixed models, rather than performing statistics on each individual timepoint. c) The dose utilized in the animal experiment (1mg/kg) is 2.5X higher than the RP2D dose and thus, the dose utilized in people (4.8mg/day vs 2mg/day),¹ which calls into question whether the results are clinically relevant. d) The tumors need to be stained for pMEK/pERK/MYC in order to determine whether the pathway identified in vitro, also exists in vivo.

7. Fig. 5G,H. The authors have a long discussion in the “Results” why they evaluated the effect of trametinib on IS tumor growth rather than on metastatic progression, indicating that the tumor growth mirrors metastatic growth. However, it is unclear why the authors didn’t evaluate the effect on metastatic growth at other organ sites, given that the cells are labeled and metastatic fluorescence can be evaluated. Since CUPs are highly metastatic, this seems to be an important evaluation for the effectiveness of trametinib. Moreover, in the discussion of Fig. 4, they indicate that organ metastases occurs early in the disease process, making it even more questionable why they didn’t assess trametinib’s effects at the ectopic sites.

8. Fig. 5I,J. The authors show that a 1,000 gene signature can predict sensitivity to trametinib; however, they never confirm the data by other means (e.g. western blot), and never use information from this signature to perform mechanistic (rescue) experiments aimed at identifying critical proteins whose expression can reverse trametinib sensitivity in agrosphere cultures.

9. Fig. S5C. Data for AS914 are missing.

10. Fig. S5D. Cell viability data for AS901, AS906, AS914 are missing even though the authors make the point that only one agnosphere responded to the drugs, indicating that they tested all of the cell lines. Also, the cell viability assays were performed at too long of timepoints (see point #5).

11. Fig. S5F. The timepoint utilized for preparing the lysates for western blots is too long (see point #5c above), and blots for the other cell lines are missing (only two agnospheres are shown). In the text (pg. 14), the authors indicate that the ERBB inhibitors are likely ineffective in preventing viability in the majority of lines (except for AS43) because the cells have found another way to activate EGFR signaling (pERK, pAKT). However, neither ERK nor AKT phosphorylation (EGFR downstream signaling) is altered in either AS43 (sensitive) or AS901 (resistant) cells following lapatinib/afatinib treatment, indicating that this isn’t a likely explanation. Critical mechanistic experiments aimed at uncovering why ERBB inhibitors are effective (or ineffective) are lacking. Also, the afatinib label is missing from the figure.

12. Fig. S5G. Trametinib viability data is only shown for two agrospheres—there rest are missing.

References:

1. Simbulan-Rosenthal CM, Dakshanamurthy S, Gaur A, Chen YS, Fang HB, Abdussamad M, et al. The repurposed anthelmintic mebendazole in combination with trametinib suppresses refractory NRASQ61K melanoma. *Oncotarget* 2017;8:12576-95

NCOMMS-20-28191

“Stem-like cells from Cancer of Unknown Primary (CUP) are endowed with distinctive hypermetastatic properties and unveil liability to MEK inhibition” by Federica Verginelli et al.

POINT-BY-POINT REPLY TO REVIEWERS' COMMENTS

Reviewer #1

General comment

In their manuscript Verginelli and coworkers present evidence suggesting that cancers of unknown primary (CUP) represent a distinct pathological entity originating from the malignant transformation of stem-like cells. For their experiments they have generated patient-derived xenografts (PDX) and spheroid cultures (agnospheres) from tumor tissue of six and five patients with CUP syndrome, respectively. Using these tools they describe that agnospheres behave as CUP-initiating, stem-like cells with cell-autonomous, early dissemination and widespread metastasis. This phenotype resulted from constitutive activation of MAP kinases and MYC, leading to sensitivity towards MEK inhibitors. They conclude that MEK inhibition might constitute a novel treatment paradigm in CUP syndrome as a transcriptomic signature predicting response towards MEK inhibitors was found in 75% of CUP patients. Although the manuscript is interesting and concerns a clearly understudied cancer type, it suffers from several shortcomings that need to be taken care of before publication.

Major Point N.1

A consistent control group, representing at least some of the major cancer types, that is stringently used throughout all experiments shown in the manuscript needs to be established. In Figure 2, only CRC729 (colorectal cancer) and MS321 (melanoma) are shown as controls in panel A, no control is displayed in panel B, only CRC729 is shown in panel C, CRC729, CRC0155 (typo in panel D) and MS321 are used in panels D - F, and no control is shown again in panel G. In contrast, the figure legend of Figure 2 lists explanations for abbreviation of glioblastoma, colorectal cancer, pancreatic cancer, melanoma and leukemia.

Major Point N.2

Along the same lines, for the experiments shown in Figures 3 - 5, with the exception of single controls in Figures 4E (CRC729) and 5A (MS321), no controls are given at all. Instead, organoids derived from aggressive metastases of known origin need to be used as comparators here.

Reply to both points N.1 and N.2

We understand the Reviewers' concerns about a 'comparison' group representing some of the major cancer types to be used throughout the experiments. As we now better highlight at page 8 of the revised manuscript, we carefully chose a group of comparator tumorspheres that (i) derive from metastatic tumors of known origin and (ii) share with agnospheres the remarkable property to propagate in the absence of exogenous growth factors, thus allowing functional assessments in the same conditions as agnospheres. These models are colospheres mCRC729 and mCRC0155 (derived from metastases of colorectal cancers) and melanosphere mMS321 (derived from a melanoma metastasis). In the original manuscript, such models were used only in some experiments. In the revised manuscript, we did our best to include these comparators in all relevant experiments and to highlight the differences (or analogies) between agnospheres and these models, as detailed below. However, please note that many experiments are aimed at characterizing agnosphere quantitative properties (e.g. clonogenic or TIC frequencies in vitro and in vivo) and cancer stem cell (CSC) prerogatives, such as regeneration of the original tumor histology, whose absolute outcomes would not change even including other tumorspheres as comparators. Moreover, in the revised manuscript we included new panels of comparator tissues in transcriptomic analyses, as detailed below.

(i) Figure 2 (agnosphere enrichment in stem-like properties).

In the new Supplementary Fig. 2c (related to Fig. 2b) we added immunofluorescences for stem cell transcription factors in mCRC729. In the new Fig. 2c we show a Western Blot including all 6 agnospheres and the 3 tumorspheres mCRC729, mCRC0155 and mMS321. We completely redraw Fig. 2d (embryonic stem cell signature, see below), including new crucial comparators such as human Embryonic Stem Cells (hESC), an extended panel of CUP original tissues (in part previously shown in Supplementary Fig. 2D), a new panel of ~1000 breast cancer tissues pooled by grades. Analyses of EMT TFs and CSC markers in agnospheres, previously shown in Fig. 2E-F have been moved in the new Supplementary Fig. 2d-e, to be directly compared with original CUP tissues.

Fig. 2d. Heatmap showing the Gene Set Enrichment analysis of the 13 gene sets included in the Embryonic Stem Cell signature performed on the transcriptome of human embryonic stem cell lines (hESC), agnospheres, tumorspheres from metastatic cancers, CUP original tissues, and a panel of breast cancer tissues, pooled by grade (the average enrichment score is shown for each grade).

(ii) **Figure 3** (tumorigenic potential). In the revised manuscript we added an *in vivo* limiting dilution assay performed with melanosphere mMS321, as melanoma is known to date as the tumor most enriched in TIC, finding a TIC frequency comparable to agnospheres (new Supplementary Fig. 3c, see below). Moreover, we better detailed results from previous studies measuring TIC frequency in tumorspheres, tumor tissues and in CSC subpopulations prospectively isolated from tumors of known origin (page 13, Refs. 30-33).

c

		mMS321				
		Outgrowths/injections				TIC frequency (C.I.)
		10 ⁴	10 ³	10 ²	10 ¹	
P0		4/4	5/5	6/6	4/7	1:11,8 (4,34 - 31,9)

Supplementary Fig. 3c. *In vivo* limiting dilution assay of metastatic melanosphere mMS321. The indicated numbers of melanosphere cells (10⁴-10¹) were transplanted subcutis into immunocompromised mice (P0). TIC: Tumor-initiating cell frequency. C.I.: confidence interval.

(iii) **Figure 4** (metastasis formation). As in previous literature there is no evidence of spontaneous metastasis after subcutaneous transplantation of human primary cells from metastatic carcinomas of known origin (Ref. 34), in the original manuscript we showed a control experiment where we carefully monitored for 7 months, by *in vivo* and *ex-vivo* imaging, the metastogenic potential of colosphere mCRC729 (derived from a liver metastasis), with negative results (Fig. 4E and Supplementary Fig. 4D-E of the original manuscript). In the revised manuscript, this experiment (Fig. 4e and Supplementary Fig. 4d-e) and evidence from the literature (Ref 34) are better highlighted at page 15-16.

(iv) Figure 5 (trametinib sensitivity in vitro and in vivo). Please note that, in the original manuscript, the golden standard positive controls for trametinib treatment, i.e. BRAF^{V600E} melanomas, were shown in Supplementary Fig. 5g (now Supplementary Fig. 5h: mMS321 treated in vitro with trametinib, with the addition of selumetinib treatment) and Fig. 5I-J (now Fig. 5g-h: trametinib response signature in mMS321 and in melanoma tissues mMEL321, MEL2 and MEL4). For further comparison, in the revised manuscript we added the assessment of the trametinib signature in a panel of early metastatic tumors (and 3 more CUP cases, new Fig. 5h, shown below). Please, see also Major point #4 for overlapping details.

Fig. 5h. Bar graphs of enrichment scores of the trametinib response signature in tumor tissues from melanomas (MEL-), CUPs (AGN-), and early metastatic cancers of known origin (H&N: head and neck, BRE: breast, MERK: merkeloma).

Major Point N.3

Figure 5: The effect of trametinib on metastases should to be evaluated in addition to its effect on the injection site tumors.

Reply

We thank the Reviewer for this suggestion. In the revised manuscript, we show the inhibitory effect of trametinib on lung metastases from spheropatiens transplanted with agnosphere AS43 (page 20, and new Supplementary Fig. 5k, shown below). Moreover, we added a new transcriptomic analysis

of 6 metastases sampled immediately *post-mortem* from patient AGN43 (page 21-22 and new Fig. 5i, shown below) indicating that different metastases of the same CUP patient may share the same trametinib sensitivity.

Supplementary Fig. 5k. Lung sections from spheropatiens transplanted with AS43, treated with control or trametinib and sacrificed at the experimental endpoint (tumor volume=1600 mm³). Left, Representative hematoxylin and eosin stainings. Arrows point to metastatic emboli. Scale bar 50µm. Right, Quantification of metastatic emboli identified in lung sections. After whole-lung sectioning and staining, the total number of metastatic emboli was counted. In each embolus, the total number of cancer cells was counted. The dot plot reports the number of cancer cells in each and every embolus identified in trametinib-treated (n=3 mice; total emboli=11; emboli/mouse =3,6) or control mice (n=1; total emboli=12). The difference between the mean±SEM cell number in each embolus in treated vs. control mice is significant (Wilcoxon test, 1 tail, *p=0,0497).

Fig. 5i. Bar graph of enrichment scores of the trametinib response signature in different metastases of CUP patient AGN43, collected at warm autopsy (n=3 fragments of the same metastasis, mean±SEM).

Major point N.4

In the abstract of their manuscript the authors state that sensitivity to MEK inhibitors “is associated with a transcriptomic signature predicting that 75% of CUP patients could be eligible to MEK inhibition”. This conclusion derives from the transcriptomic analysis of paraffin-embedded primary tumor tissue from only 12 CUP cases (including those matched to the four agnospheres used in the manuscript) and without any patient having been treated with trametinib (for example on a compassionate use basis), and seems therefore clearly overstated.

Reply

We agree that our cohort may seem small, as the study was retrospectively performed on the remaining of bioptic samples, often scanty for the difficulties in getting biopsies from such patients. Moreover, patients were defined as CUP by stringent criteria, after a thorough, ‘ad excludendum’

immunohistochemical protocol (exemplified in Supplementary Table 1). This implied that less than 50% (27/61) of suspected CUP patients enrolled in our clinical protocol was confirmed as a ‘true CUP’ at the end of the diagnostic workout (as stated at page 6 of the revised manuscript). To better support our conclusions about eligibility to trametinib treatment, we could analyze the trametinib signature in 3 newly diagnosed CUP cases, added to the new Fig. 5h (shown above, see Reply to Major Point N.2). Overall, 11/15 CUPs were predicted to be sensitive by the transcriptomic signature, corresponding to 73.3%. We have corrected the percentage in the abstract from 75% to ‘more than 70%’. We hope that the importance of trametinib sensitivity is also corroborated by the evidence presented in the new Fig. 5i (see Reply to Major point N.3), showing that 6 metastases sampled post-mortem from a patient that underwent multiple lines of treatment are invariably sensitive to trametinib.

Moreover, in the analysis of the trametinib response signature, we’ve also added as comparator a set of ‘early metastatic patients’ (n = 10), enrolled in our clinical protocol as ‘suspected CUPs’ for their clinical presentation, but whose tumor of origin was subsequently identified based on immunohistochemical analysis (new Fig. 5h, shown above, see Reply to Major Point N.2). Interestingly, in this subset, trametinib sensitivity is predicted by the signature only in 3/10 cases. Concerning CUP patient treatment with trametinib, based on the evidence provided by this study we planned a proof-of-concept trial with 6 CUP patients that failed first-line chemotherapy. So far, we cannot report of any CUP patient treated with trametinib on compassionate basis, as, in Italy, in the absence of the BRAF mutation biomarker, such treatment is issued only to patients that exhausted multiple chemo and radio-therapy lines, and therefore are often unfit to receive the drug long enough to adequately monitor a therapeutic response.

Major Point N.5

Page 8: The authors describe that EMT transcription factor expression was relatively weak in adeno-CUPs but high in CUPs derived from carcinomas with sarcomatoid features. This finding obviously argues against a common CUP pathophysiology and phenotype and against a major role of the EMT for metastasis in CUP as the vast majority of CUPs are adenocarcinomas without displaying sarcomatoid features.

Reply

We fully agree with this Reviewer that EMT constitutive activation is not a CUP common feature, as we observed it only in the subgroup with sarcomatoid features and not in the majority of CUPs, displaying adenocarcinoma features. We have further elaborated on this point at page 10-11 of the revised manuscript. Indeed, evidence presented in this study indicates that CUPs tend to share cell-

autonomous proliferative and self-renewal properties in vitro and high tumorigenic and metastogenic potential in vivo, rather than any other genetic or phenotypic feature, such as EMT constitutive activation. However, we would like to speculate that, as EMT is well recognized as a reversible process, and mesenchymal-epithelial transition seems required for efficient metastatization, we cannot exclude that agnospheres from adenocarcinomas can be induced to undergo EMT by appropriate exogenous factors. CUP EMT in response to tumor microenvironmental factors is under evaluation in a separate study.

Minor comment N.1

The manuscript needs language editing.

Reply. The manuscript has been carefully edited.

Minor comment N.2

On page 6 it is stated that some of the agnospheres harbored mutations “some of which are among the ten most frequently mutated genes in CUPs, being enriched in CUPs compared with metastases from cancers of known origin”. This suggests that CUP-specific mutational spectra exist, which is not the case.

Reply

The statement at page 6 is based on Supplementary Fig. 1, reporting the frequency of the top 10 mutated genes in CUPs (n = 146) and metastases of known origin (n = 4595) in the MSK-IMPACT Clinical Sequencing Cohort (http://www.cbioportal.org/study/summary?id=msk_impact_2017). To our knowledge, this is the only publically available mutational dataset that includes both CUPs and metastases from tumors of known origin. These data were shown not to suggest CUP-specific mutational spectra, but simply to compare mutations found in our small CUP cohort with an independent large patient cohort. Moreover, to show these data seems coherent with the Reviewers' request to display more comparison groups (metastases from tumors of known origin). However, if the Reviewer still thinks that data presented in Supplementary Fig. 1 are misleading we can delete it without detracting any substantial evidence from the manuscript.

Reviewer #2

General comment

This study addresses an important issue in the cancer field, which is modeling, characterizing and finding therapeutic targets for Cancer of Unknown Primary (CUP). This is one of the few groups in the world that have addressed this intractable but actually quite prevalent type of cancer diagnosis so it is a meritorious effort that may have significant clinical impact. (...)

The most interesting findings of this work come from the transplantation of agnospheres in mice. As few as 10 cells were able to generate subcutaneous tumors, indicating a potent tumor-initiating ability of agnosphere-derived cells. Importantly, the established tumors faithfully recapitulated early dissemination and metastatic spread to multiple organs, as seen in CUP patients. The authors further found that growth was dependent on MEK1/2 and agnospheres were sensitive to MEK1/2 inhibition via Trametinib treatment both in vitro and in vivo. Overall, the study is innovative in the field and provides a first and powerful model to gain insight into the biology and therapeutic approaches to target CUP. Some technical concerns and further experiments to strengthen the findings are listed below.

Reply

We are deeply grateful to, and motivated by, this Reviewer for his/her flattering appreciation of our pioneering effort to build a faithful CUP disease model, and his/her consideration of the potential clinical impact of our mechanistic findings.

Major Point N.1

(i) *In Figure 2, the authors provide RT-qPCR, Western blotting, Immunofluorescence and RNA-seq data to support the hypothesis that agnospheres express stem cell markers and factors. However, in panels A-C, it would be more convincing to include a positive control and a negative control (e.g. a tissue-derived or in vitro induced stem cell as positive and another cancer cell – patient-derived or cell line – as negative).*

Reply

Findings in Fig. 2, reported in the paragraph entitled ‘Agnospheres are enriched in stem-like cells that self-sustain their long-term propagation’, are aimed at characterizing the stem features of agnospheres and CUP tissues in absolute terms. Moreover, technical negative controls are intrinsic in the experiments (new Fig. 2a-c), as, for each tested factor, different expression levels or negative samples can be observed within the panel. However, as suggested, we did our best to compare

agnospheres with other spheres from tumors of known origin, in the new Fig. 2 and throughout the revised manuscript.

As we now better highlight at page 8 of the revised manuscript, we carefully chose a group of comparator tumorspheres that (i) derive from metastatic tumors of known origin and (ii) share with agnospheres the remarkable property to propagate in the absence of exogenous growth factors, thus allowing functional assessments in the same conditions as agnospheres. These models are colospheres mCRC729 and mCRC0155 (derived from liver metastases of colorectal cancers) and melanosphere mMS321 (derived from a melanoma metastasis). In the original manuscript, such models were used only in some experiments, while in the revised manuscript we tried to add them in all relevant experiments, and to highlight the differences (or analogies) between agnospheres and these models throughout the manuscript. In particular, in the Western blot shown in the revised Fig. 2c (see below), beside 2 new agnospheres we added tumorspheres mCRC0155 and mMS321, and in the immunofluorescence analysis shown in the new Supplementary Fig. 2c (see below) we added mCRC729.

Fig. 2c. Western blot analysis of Polycomb repressor factors, reprogramming and EMT TFs, and epithelial and mesenchymal markers ($n \geq 3$) in agnospheres and tumorspheres mCRC729, mCRC0155 and mMS321 .

Supplementary Fig. 2c. Immunofluorescence staining in colosphere mCRC729 for the indicated markers. Single split channels are shown for triple labelling. Scale bar, 50µm. Inset: magnification of the dotted area. Scale bar, 10µm.

(ii) The data in D and E is displayed in a confusing way. It would be useful to see how the signatures and genes compare to ESC, rather than having exclusively absolute values and scores

from the RNA-seq analysis. Most importantly, the media used for agnosphere formation naturally enriches for stem cells, as there are selected to survive in this condition. Thus, detection of these markers from fresh tumors would be helpful. Alternatively, qPCR from RNA from FFPE samples or IF on the sections may be a good validation of the markers in the patient samples. Having other less aggressive type of cancer cells in the same media would provide an insightful comparison to rule out the argument that what is seen is not specific to CUP agnospheres but due to the culture conditions. For example tumor from the same tissue that did not present as CUP. Agnospheres could also be dissociated, removing the cells from stem-cell media and placing them in regular media. Do they still display/retain stem-like properties?

Reply

We thank the Reviewer for this remark that compelled us to fully redraw Fig. 2D-F and related Supplementary Fig. 2D-E. In the revised Fig. 2d (expression of the embryonic stem cell signature, shown below), as suggested, we added the analysis of 4 lines of human Embryonic Stem Cells (hESC), an extended panel of CUP original tissues (in part shown in the original Supplementary Fig. 2D), and a new panel of ~1000 breast cancer tissues pooled by grades. Transcriptomic analyses of EMT TFs and CSC markers in agnospheres, previously shown in Fig. 2E-F have been moved in the new Supplementary Fig. 2d-e, to be directly compared with original CUP tissues. Please note (as now better highlighted in the text, page 9) that while, in ES cells, maintenance of the transcriptional program sustaining the ES status requires exogenous factors (Ref. 19 and 20), in agnospheres most of the same program is driven in a fully cell-autonomous manner.

Fig. 2d. Heatmap showing the Gene Set Enrichment analysis of the 13 gene sets included in the Embryonic Stem Cell signature performed on the transcriptome of human embryonic stem cell lines (hESC), agnospheres, tumorspheres from metastatic cancers, CUP original tissues, and a panel of breast cancer tissues, pooled by grade (the average enrichment score is shown for each grade).

Another main conclusion that can be drawn from this direct comparison of ES cells, agnospheres, CUP tissues and various breast cancer grades, is that CUP tissues are particularly enriched in cells

with stem-like features, which is mirrored by the high frequency of tumor-initiating cells in agnospheres. Moreover, in the revised manuscript (page 13), we better highlighted that expression of Polycomb repressors (BMI1 and EZH2) in original CUP tissues and in tumors derived from agnosphere injection (Fig. 3B, see below), overlaps with expression of the same proteins in agnospheres (Fig. 2B). This indicates that agnospheres retain (embryonic) stem properties already present in the original tumors.

Fig. 3b. Immunofluorescent staining of Polycomb repressors in CUP patient metastases (left columns) and tumors formed by agnospheres at the subcutaneous injection site (right columns). Nuclei were counterstained with DAPI. Scale bar, 50 μ m.

Major Point N.2

(i) *In Figure 5, the authors explore the effect of Trametinib on agnospheres in culture and in the CUP preclinical model. ATP consumption should be measured at the same timepoint for all agnospheres (some are measured at 4 days, others at 10 days).*

Reply

In Fig. 5, the effect of trametinib is shown at different time-points (4 or 10 days) based on the following parameters: (i) the different agnosphere cell doubling times (2 days for AS901, AS906 and AS67, 3.5 days for AS43, 5 days for AS914, and 10 days for AS47, as reported at page 9 of the original manuscript and updated at page 11 of the revised text); (ii) the time required to detect a full cytotoxic effect (if any), as established in preliminary time-course experiments. In particular, 10 days were required to observe a full cytotoxic effect in AS43. As shown in the new Supplementary Fig. 5f (see below), in a dose-response experiment at 4 days, AS43 response to trametinib was incomplete, while, as shown in the original manuscript (Fig. 5C), after 10 days the response was complete.

Please note that, in the case of trametinib-resistant agnospheres, viability at the longest time-point was unchanged or paradoxically increased, indicating lack of unspecific toxic effects.

f

Supplementary Fig. 5f. Cell viability measured in agnospheres treated for 4 days with the indicated doses of trametinib, selumetinib or vehicle (DMSO) through quantification of relative ATP consumption normalized vs. untreated cells (NT) (trametinib: AS901 n=4, AS906 and AS43 n=3, and selumetinib: AS901 n=4, AS906 and AS43 n=5, mean±SEM).

(ii) As shown in Supplementary Figure 5, AS914 agnospheres had increased proliferation in response to Trametinib, in contrast to the other patients' spheres. From the characterization and RNA-seq, the authors should attempt to find what is specific to that sample compared to the others to provide an explanation for this result.

Reply

The study of mechanisms that explain primary resistance to trametinib, and even paradoxical responses, is indeed very important, and RNAseq can provide essential hints. However, as discussed by the literature (Ref. 37), and mentioned in our Discussion (page 25-26), paradoxical responses to trametinib, as observed in AS914, are not surprising and the underlying mechanisms are likely multifaceted. We are actively pursuing these investigations on the entire CUP panel with the aim of finding why most CUPs are sensitive even in the absence of BRAF mutations, and why the remaining are resistant. We feel that disclosing data on specific cases such as AS914 could be premature. However, as shown in the new Fig. 5c (see below), a common feature of agnospheres insensitive to trametinib is lack of MYC downregulation in response to the drug. Therefore, we can anticipate that mechanisms keeping MYC expression levels high in spite of MEK inhibition are candidates to explain resistance and paradoxical responses.

Fig. 5c. Representative Western blot analysis of agnospheres treated 48 h (AS901, AS906 and AS67) or 96 h (AS43, AS914 and N-AS47) with vehicle (-) or trametinib 50 nM (+) (p-: phosphorylated; cl. PARP: cleaved PARP.) (n=3).

(iii) Other Mek inhibitors should be tested or genetic approaches to KD Mek and ERK should be employed to explain the differences.

Reply

As suggested, we assessed a second FDA-approved MEK inhibitor, selumetinib, known to be highly selective for MEK1/2, but less effective than trametinib in preventing MEK reactivation by RAF. As shown in the revised Supplementary Fig. 5f-i (see below), and discussed in the revised text (pages 17-18), the effects of selumetinib, although weaker, as expected, were comparable to trametinib. In particular, AS914 responded with paradoxical hyperproliferation to both inhibitors.

f

h

i

Supplementary Fig. 5. **f**, Cell viability measured in agnospheres treated for 4 days with the indicated doses of trametinib, selumetinib or vehicle (DMSO) through quantification of relative ATP consumption normalized vs. untreated cells (NT) (trametinib: AS901 n=4, AS906 and AS43 n=3, and selumetinib: AS901 n=4, AS906 and AS43 n=5, mean±SEM). **h-i**, Cell viability of melanosphere mMS321 (**h**) and agnospheres AS67 and AS914 (**i**), treated with selumetinib (500 nM), or trametinib (50 nM), or vehicle (DMSO), measured through quantification of relative ATP consumption normalized vs. untreated condition (NT) at day 10 (mMS321: n=3, AS67: n=4, AS914 n=3, ANOVA, Bonferroni multicomparison test: selumetinib or trametinib vs DMSO, *** p ≤ 0.0001).

(iv) Mutational analysis of the Ras pathway should also be provided to rule out that as in AS47 line the lack of sensitivity is not due to a super-active Ras mutant pathway.

Reply

WES mutational analysis of AS47 was provided in the original manuscript (Datasheet 1, now accession code EGAD00001006668) and described at page 6: N-AS47 does not harbor any mutation in known oncogenes or tumor suppressor genes, and, as only gene copy number variation,

it harbors heterozygous PTEN loss. Super-activation of the RAS pathway should therefore be excluded. We could correlate trametinib resistance with persistence of N-MYC expression, which can vicariate for c-MYC.

Major Point N.3

To assess Trametinib effect in vivo, they monitored the response on subcutaneous tumors. Although the authors provide multiple explanations for this choice and the results are convincing, it is necessary to look at secondary organs. CUPs do not present as a primary and the multiorgan metastasis patterns may also reveal differenced in the response to MEK1/2 inhibition that could be highly informative for potential clinical trials. Show tumor volume change until end of treatment. Further, to reproduce what seen in CUP patients, the authors should first remove the primary tumor by surgery, then starting Trametinib treatment and evaluating the effect on metastatic outgrowth. Also since in some animals dissemination occurs almost immediately after implantation it could be tested if MEK1/2 is also needed for this process as it may also provide insight into how this pathway is wired for dissemination and growth in these tumors.

Reply

(i) As suggested, in the revised manuscript, we show the inhibitory effect of trametinib on lung metastases from spheropatient transplanted with agnosphere AS43 (page 20, and new Supplementary Fig. 5k, shown below). Moreover, we added a new transcriptomic analysis of 6 metastases sampled immediately post-mortem from the human patient AGN43 (page 21-22 and new Fig. 5i, shown below), indicating that different metastases of the same CUP patient may share the same trametinib sensitivity. Of note, the patient underwent multiple lines of chemotherapy that could exert important selective pressures. This is a good omen for therapeutic effectiveness in a proof-of-concept clinical trial that we are planning, where patients will be treated with trametinib after a first-line chemotherapy, and, for analysis of the trametinib signature, may undergo analysis of a single, chemo-naïve metastasis at the time of diagnosis.

Supplementary Fig. 5k. Lung sections from spheropatients transplanted with AS43, treated with control or trametinib and sacrificed at the experimental endpoint (tumor volume=1600 mm³). Left, Representative hematoxylin and eosin stainings. Arrows point to metastatic emboli. Scale bar 50 μm. Right, Quantification of metastatic emboli identified in lung sections. After whole-lung sectioning and staining, the total number of metastatic emboli was counted. In each embolus, the total number of cancer cells was counted. The dot plot reports the number of cancer cells in each and every embolus identified in trametinib-treated (n=3 mice; total emboli=11; emboli/mouse =3,6) or control mice (n=1; total emboli=12). The difference between the mean±SEM cell number in each embolus in treated vs. control mice is significant (Wilcoxon test, 1 tail, *p=0,0497).

Fig. 5i. Bar graph of enrichment scores of the trametinib response signature in different metastases of CUP patient AGN43, collected at warm autopsy (n=3 fragments of the same metastasis, mean±SEM).

(ii) Concerning the request of showing tumor volume fold change until the end of the experiment, please note that, in the original manuscript, we chose to present these data as a survival curve, showing for each mouse the time required to reach the experimental endpoint (tumor volume = 1600 mm³, Fig. 5E). We think that graphical representation of tumor volume fold change until the end is misleading: as mice reach the experimental endpoint (and thus, they are excluded by mean and SEM calculation), curves lose linearity and standard deviations increase, even though statistical significance is kept throughout the experiment, as assessed, in the revised manuscript, by two-way ANOVA (new Fig 5d). In addition, raw data and statistical analysis with linear mixed model of absolute tumor volume curves are now provided in the Source Data file.

Major Point N.4

Although Trametinib is an interesting candidate compound from this analysis, an alternative therapeutic strategy would make the work stronger (as potential acquired resistance has not been addressed in detail and serious side effects have been shown in patients). Going back to the RNA-seq to identify novel targets specific to CUP samples would bring a more innovative and impactful

side to the study. For example WBs in Figures 2 and 5 offer other alternative treatments such as inhibitors or mAbs targeting surface receptors such as EGFR, HER2 IGFR and Met. These are heterogeneously activated in the different agnospheres but not much explanation is provided as to why these are different. Also two agnospheres including a MEK1/2 inhibitor-resistant tumor seem to upregulate EZH2 and there are excellent therapeutics for this chromatin remodeler. It seems that as CUP is very diverse their mechanism of growth promotion may be diverse and limiting it to Mek1/2 inhibition may be less effective. Testing these other targets with drugs that are available may be useful.

Reply

As stated in the original manuscript (page 14), we choose to target MEK1/2 with trametinib as the MAP kinase pathway is the cross-road of multiple tyrosine receptor autocrine loops that, as noted by this Reviewer, are heterogeneously activated in the different agnospheres and, likely working in different combinations in each case, can sustain the keen proliferative autonomy of agnospheres. As shown in Fig. 5B of the original manuscript, and better documented in the new Fig. 5a (Western blot of growth factor receptors in all agnospheres and tumorspheres), a combination of active (phosphorylated) EGFR family members is often present. Therefore, we attempted treatment with the pan-EGFR family inhibitors lapatinib or afatinib in representative agnospheres with negligible effect in AS901 and proliferative inhibition without induction of cell death in AS43 (Supplementary Fig. 5D-F of the original manuscript). In the revised manuscript, we have shown also the negative outcome of pan-EGFR inhibitors in AS906, where HRAS mutation is expected to confer primary resistance to tyrosine kinase receptor inhibition (new Fig S5d-e).

As several receptors seem simultaneously activated in agnospheres, a combination with multiple inhibitors (and/or antibodies) should be assessed, which however seems poorly translatable to the clinics for the high risk of toxicity. This may be circumvented by the use of trametinib, which is predicted to be effective in 70% of CUP patients based on the transcriptional signature.

We thank the Reviewer for suggesting other targets that could be active in cases insensitive to trametinib, such as chromatin remodelers. This important issue will be investigated in such specific cases but it seems beyond the scope of this paper.

Minor Points

Minor Point N.1

In Figure 2, please include insets of immunofluorescence images, particularly where the arrows are pointing to co-localization.

Reply. We thank the Reviewer for this suggestion. Figure 2b has been modified accordingly.

Minor Point N.2

Figure 3 demonstrates the high tumorigenic capacity of agnospheres, with an elegant in vivo limiting dilution transplantation experiment. I suggest to include in the experiment a control of spheres generated from other cancer cells (as shown later for melanospheres and colospheres) from the same tissue of origin but not derived from CUP.

Reply

In the revised manuscript we added an in vivo limiting dilution assay performed with melanosphere mMS321 (derived from an intestinal melanoma metastasis). To date, melanoma is known as the tumor most enriched in TIC, and we found that its TIC frequency is comparable to agnospheres'. (Page 13 and Supplementary Fig. 3c, see below). Moreover, we better detailed results from previous studies measuring TIC frequency in tumorspheres, tumor tissues and in CSC subpopulations prospectively isolated from tumors of known origin (page 13, Refs. 30-33).

c

mMS321				
Outgrowths/injections				TIC frequency (C.I.)
10 ⁴	10 ³	10 ²	10 ¹	
P0 4/4	5/5	6/6	4/7	1:11,8 (4,34 - 31,9)

Supplementary Fig. 3c. In vivo limiting dilution assay of metastatic melanospheres mMS321. The indicated numbers of melanosphere cells (10⁴-10¹) were transplanted subcutis into immunocompromised mice (P0). TIC: Tumor-initiating cell frequency. C.I.: confidence interval.

Minor Point N.3

In Figure 5 and Supplementary Figure 5, relative ATP consumption by Cell Titer Glo kit is used to assess agnosphere viability. Agnospheres were dissociated and seeded as single cells in the experiment. Cell viability should also be measured by treatment of agnosphere cells in full, non-dissociated spheres to establish whether trametinib has the same effect.

Reply

We appreciate this suggestion and, indeed, in set-up experiments we also assessed treatment of whole agnospheres. However, in this setting, results were difficult to interpret, likely due to heterogeneity of spheroid dimensions in technical replicates of agnosphere cultures. We observed reproducible results only by seeding and treating agnosphere replicates formed by equal numbers of single cells.

Minor Point N.4

A genetic proof of MEK dependency in agnospheres must be used to strengthen what was found with pharmacological Trametinib treatment. Using siRNA or CRISPR to eliminate MEK and ERK in the case of AS47 in these cells will show specificity and selectivity.

We understand this point, but, as an alternative to genetic inactivation (and as suggested by this Reviewer in Major Point N.2), in the revised version we added experiments performed with a second FDA-approved MEK1/2 inhibitor (selumetinib), showing overlapping results with trametinib in all agnospheres (see above, Reply to Major Point N.2, part iii, Supplementary Fig. 5f-i). We thus hope this is sufficient to demonstrate that MEK1/2 is key to support the CUP phenotype.

Minor Point N.5

On pages 12/13, in the Results section, the authors state that “single disseminated cells were identified by pan-cytokeratin immunostaining, thus retaining their epithelioid phenotype, suggesting that EMT, if ever occurred, rapidly disappeared”. I suggest rephrasing this sentence as expression of pan-cytokeratin does not exclude that these cells are still residing in a partial EMT state.

Reply

We thank this Reviewer for noticing. We revised the sentence accordingly (page 15).

Minor point N.6

Further below in the text, the authors claim that the inability to form detectable metastases from colospheres injected subcutaneously “is consistent with lack of evidence in similar models reported by the literature”.

Reply. The paragraph has been revised and a reference (N. 34) has been added (page 15).

Reviewer #3

General comment

(i) *Strengths of this manuscript include the significance of studying this class of patients who have few treatment options. In addition, the manuscript includes a plethora of sequencing/bioinformatics analyses, which provide a good foundation for mechanistic studies. Moreover, the authors utilize in vitro and in vivo experiments based on patient-derived material to evaluate their hypotheses. Furthermore, most experiments are well-controlled, data are clear, the manuscript is well-written, and statistical measures are robust, in most instances.*

Reply. We thank this Reviewer for his/her appreciation of our study.

(ii) *However, despite these strengths, the manuscript also contains a number of weaknesses, which detract from the strengths. Firstly, although the authors perform Whole Exome sequencing on the agnospheres, they never link genomic alterations with pathway activation identified in subsequent in vitro and in vivo studies.*

Reply

In the revised manuscript, we strived to better discuss the relationship between the (few) genetic alterations found in agnospheres and their possible effect on cell-autonomous proliferation. At page 7 we mention that, in our previous studies, cancer stem cells derived from other aggressive tumors such as glioblastomas or colorectal cancer metastases could be in vitro long-term propagated in the absence of exogenous growth factors, like agnospheres, only in rare cases, mostly featuring constitutive activation of the growth factor receptor/RAS pathway (Ref. 15-17). Moreover, at the beginning of the paragraph entitled ‘Agnospheres are sensitive to MEK inhibition in vitro’ (page 16), we added a discussion of the possible relationship between genetic alterations found in agnospheres on the one hand, and their properties on the other. As found by WES analysis, AS906 harbored HRAS mutation and AS914 KRAS mutation; in the other agnospheres, genetic alterations unlikely correlated with constitutive activation of cell-autonomous proliferation. Receptor tyrosine kinases that we found phosphorylated in agnospheres were genetically unaltered (now specified at page 16). Through phospho-RTK proteomic profiling (not shown) we also found that PDGFRb, altered in hypermutated AS901, is not phosphorylated. In the RNAseq analysis we found that, in AS43, NTRK, although mutated, is not expressed (and of course not phosphorylated in the phosphoproteome), as now specified in the revised Supplementary Table 3. Therefore, we can conclude that proliferative autonomy of agnospheres lacking RAS family mutations can be explained by expression of several growth factors, accompanied by expression and phosphorylation of their cognate receptors, indicating the occurrence of autocrine loops (page 16, Fig. 5a and Supplementary Fig. 5b). In all agnospheres, either in case of RAS mutations or functional activation of unaltered growth factor receptors, the MEK/ERK/MYC pathway appears highly activated.

(iii) *Secondly, much of the data shown are merely correlative. The mechanistic data that the authors provide do not adequately or definitively link the MEK/ERK/MYC pathway to CUP proliferation/survival, stem cell features and trametinib sensitivity as important rescue experiments are lacking, and the authors do not provide corroborating evidence for the importance of this*

pathway in primary patient samples (uncultured; “original tissues”) and in animal tumors from the in vivo studies.

Reply

In the original manuscript we showed that mechanisms linking MEK inhibition by trametinib and the agnosphere proliferative block and apoptosis involve (i) full shut-off of well-known main cell cycle regulators, namely expression of MYC and its transcriptional target cyclin D, and pRB phosphorylation, accompanied by upregulation of the cell-cycle inhibitor p27-KIP1 (Fig. 5D) and (ii) accumulation of cleaved PARP, a prominent marker of apoptosis (Fig. 5D). In the revised manuscript we corroborated these findings, by adding two new agnospheres, one sensitive (AS67), the other responding to trametinib with a paradoxical proliferative increase (AS914). As shown in the new Fig. 5c (shown below), this extended agnosphere panel allows to significantly correlate sensitivity to trametinib with inhibition of MYC, cyclin D and phosphoRB, and cleaved PARP accumulation. In contrast, trametinib resistance (N-AS47) or paradoxical response (AS914) correlate with lack of MYC/cyclinD/pRB downregulation and lack of cleaved PARP accumulation.

Fig. 5c. Representative Western blot analysis of agnospheres treated 48 h (AS901, AS906 and AS67) or 96 h (AS43, AS914 and N-AS47) with vehicle (-) or trametinib 50 nM (+) (p-: phosphorylated; cl. PARP: cleaved PARP.) (n=3).

To corroborate the importance of MEK activity, as an alternative to rescue experiments, which are quite laborious to be performed in spheroids, we carried out experiments with an alternative, FDA-approved MEK1/2 inhibitor, selumetinib, known to be highly selective for MEK1/2, but less effective than trametinib in preventing MEK reactivation by RAF. As shown in the new Supplementary Fig. 5f-i (see below) the effects of selumetinib, although weaker, were comparable to trametinib. In particular, AS914 responded with paradoxical hyperproliferation to both inhibitors.

f**h****i**
Supplementary Fig. 5. **f**, Cell viability measured in agnospheres treated for 4 days with the indicated doses of trametinib, selumetinib or vehicle (DMSO) through quantification of relative ATP consumption normalized vs. untreated cells (NT) (trametinib: AS901 n=4, AS906 and AS43 n=3, and selumetinib: AS901 n=4, AS906 and AS43 n=5, mean±SEM). **h-i**, Cell viability of melanosphere mMS321 (**h**) and agnospheres AS67 and AS914 (**i**), treated with selumetinib (500 nM), or trametinib (50 nM), or vehicle (DMSO), measured through quantification of relative ATP consumption normalized vs. untreated condition (NT) at day 10 (mMS321: n=3, AS67: n=4, AS914 n=3, ANOVA, Bonferroni multicomparison test: selumetinib or trametinib vs DMSO, *** p ≤ 0.0001).

Concerning the analysis of the MEK/ERK/MYC pathway in original CUP tissues, please note that analysis of the Embryonic Stem Cell Signature, shown in the original Supplementary Fig. 2D, indicated high expression of MYC transcription factor and its target genes in CUP tissues as well as in agnospheres (Fig. 2D). In the revised manuscript, this analysis has been extended to additional CUP cases, and shown in main Figure 2d (see below)

d
Fig. 2d. Heatmap showing the Gene Set Enrichment analysis of the 13 gene sets included in the Embryonic Stem Cell signature performed on the transcriptome of human embryonic stem cell lines (hESC), agnospheres, tumorspheres from metastatic cancers, CUP original tissues, and a panel of breast cancer tissues, pooled by grade (the average enrichment score is shown for each grade).

Moreover, to show that trametinib treatment of experimental tumors, generated by agnosphere transplantation, causes loss of ERK/MAP kinase phosphorylation, in the revised manuscript we've added the analysis of phospho-ERK1/2 in tumor sections (new Fig. 5g, see below). This analysis shows that the scanty residual areas of tumors treated with trametinib still respond to the drug with inhibition of ERK1/2 phosphorylation.

Fig. 5f. Representative H&E and immunohistochemical staining for Cyclin D1 and phospho-ERK1/2 of whole tumor sections derived from endpoint mice transplanted with AS906 and treated with control or trametinib (n=3).

(iv) *In addition, the in vitro experiments utilize long timepoints (48-96h for blots) and relatively high trametinib doses (50nM), which makes it difficult to assess a direct and specific effect on signaling pathway.*

Reply

The trametinib dose of 50 nM was chosen taking in account:

- (i) The Sanger Institute analysis of cell line trametinib sensitivity, showing that the median trametinib IC50 for BRAF-mutated melanoma cell lines (n = 25) is 36 nM, while the median IC50 for BRAF wild-type melanoma cell lines is 248 nM (see Figure R1 below).
- (ii) Our dose-response (10 nM-5µM) experiments (Fig. 5C of the original manuscript, now Supplementary Fig. 5f), showing that trametinib 50 nM elicited a maximal effect in the most sensitive cell lines (AS901 and AS906, both BRAF wild-type), as well as in BRAF-mutated mMS321. This dose concomitantly allowed to detect resistance (N-AS47) and even a paradoxical proliferative response (AS914), ruling out the occurrence of unspecific toxic effects (Supplementary Fig. 5G of the original manuscript, now S5h).

Tissue specific analysis

Skin cutaneous melanoma

Trametinib IC50 values for BRAF_mut

Click on circles to link to cell line information

Screening concentration: 0.0010005 (lower brown line) - 1.0000 (upper brown line)

	BRAF_mut	Wild type	MWW p value	Signif	Selected
Number of cell lines	25	6			
Median	0.036051	0.24800			
Geometric mean (red line)	0.036835	0.21437			
Selected groups			0.104	-	
SKCM	25	6	0.10416	-	[x]

Check all Clear all

Figure R1. Trametinib IC50 values in BRAF-mut and BRAF-wild type skin cutaneous melanoma cell lines (www.cancerrxgene.org/compound/Trametinib/1372/scatter/BRAF_mut?tissue=SKCM&screening_set=GDSC2).

Concerning timing, the effect of trametinib on signal transduction was measured at long time-points to assess whether chronic MEK1/2 inhibition resulted in consistent long-term repression of downstream master regulators of the proliferative pathway, such as MYC, phospho-RB and Cyclin D, and accumulation of cleaved PARP, in sensitive agnospheres (new Fig. 5c, see above, Reply to General Comment, part iii). However, we agree with this Reviewer that it is also important to show the trametinib effect at earlier time-points, thus we added a Western blot shown in the new Supplementary Fig. 5g (see below).

Supplementary Fig. 5g. Representative Western blot analysis of AS906 treated 30 min with selumetinib (500nM), trametinib (50nM) or vehicle.

(v) *Furthermore, the trametinib dose utilized for in vivo studies (1mg/kg), although used in some published studies, translates to 4.8mg/day in humans,¹ which is significantly above the Recommended Phase 2 Dose and thus, the dose utilized in the clinic (2mg/day), which may reduce the clinical impact of the study if this dose cannot be established in patients.*

Reply

As reported in the original manuscript (pages 19 and 40, in the revised text), we performed trametinib treatments based on a seminal reference study that characterised trametinib pharmacokinetics and established *in vivo* sensitivity of tumors generated by transplantation of cell lines harboring BRAF mutations (Ref. 41). In this study, the therapeutic range was 0.3-3 mg/kg/die, and the manufacturer's instruction (Sellekchem) for *in vivo* trametinib treatment recommends to use 1 mg/kg/die (based on Ref. 41 and Yamaguchi et al. Int. J. Oncol. 39:23, 2011, DOI: 10.3892/ijo.2011.1015). We fully agree that further studies are required to establish whether the phase II-trial recommended dose can be effective in CUP patients. However, we'd like to point out that CUP-tissue enrichment scores in the trametinib response signature are similar to those of BRAF-mutated melanoma patients that are clinically treated with 2 mg/day (Fig. 5j). Based on our preclinical results, we are setting-up a trial in CUP patients that uses the same trametinib dosage as in melanomas.

(vi) *Finally, although the investigators make the argument that the IS tumors that are generated mirror the growth of metastatic cells, the effect of trametinib on colonization of other organ sites (in vivo metastatic progression) and stem cell expansion (in vitro or in vivo), which the authors argue are critical characteristics of CUPs, is never assessed.*

Reply

In the revised manuscript, we show the inhibitory effect of trametinib on lung metastases from spheropatient transplanted with agnosphere AS43 (page 20 and new Fig. 5k, shown below; see also Reply to Point N. 7). Moreover, we added a new transcriptomic analysis of 6 metastases sampled immediately post-mortem from patient AGN43 (page 21-22 and new Fig. 5i, shown below) indicating that different metastases of the same CUP patient may all display the same trametinib sensitivity.

Supplementary Fig. 5k. Lung sections from spheropatient transplanted with AS43, treated with control or trametinib and sacrificed at the experimental endpoint (tumor volume=1600 mm³). Left, Representative hematoxylin and eosin

stainings. Arrows point to metastatic emboli. Scale bar 50 μm . Right, Quantification of metastatic emboli identified in lung sections. After whole-lung sectioning and staining, the total number of metastatic emboli was counted. In each embolus, the total number of cancer cells was counted. The dot plot reports the number of cancer cells in each and every embolus identified in trametinib-treated (n=3 mice; total emboli=11; emboli/mouse =3,6) or control mice (n=1; total emboli=12). The difference between the mean \pm SEM cell number in each embolus in treated vs. control mice is significant (Wilcoxon test, 1 tail, *p=0,0497).

Fig. 5i. Bar graph of enrichment scores of the trametinib response signature in different metastases of CUP patient AGN43, collected at warm autopsy (n=3 fragments of the same metastasis, mean \pm SEM).

Additional details and comments

Point N.1

The data are incomplete. Some figures show data for 6 agnosphere cultures, while others show 4, 3 or 2 lines. The authors need to be consistent and show all of the data either within the figure or as Supplementary data. See additional details below.

Reply. We've now added in vitro data for all 6 agnospheres in main or supplementary figures.

Point N.2

Fig. 2C. AS67 and AS914 are missing from the blots. Moreover, N-AS47, the neuroendocrine tumor which displays non-metastatic features, harbors high levels of N-MYC which is contrary to the authors' hypothesis that MYC signaling drives aggressive CUP disease.

Reply

AS67 and AS914 have been added to in the revised Fig 2c (shown below).

Concerning the comment on N-AS47, please note that we didn't imply that aggressiveness (meant as quick metastatic spread) is directly proportional to the levels of either c-MYC or N-MYC. Rather, we correlated high levels of MYC expression, sustained in all agnosphere in the absence of exogenous factors, with the ability to grow in a very autonomous manner, a property that can be key to the ability to metastasize to multiple organs. Please note that N-AS47 do metastasize, although at later timepoints compared with the others (Fig 4E-F of the original manuscript). Moreover, high levels of N-MYC expression tightly associate with the neuroectodermal origin of N-AS47, as mentioned in the Discussion (page 24, Ref. 51).

Fig. 2c. Western blot analysis of Polycomb repressor factors, reprogramming and EMT TFs, and epithelial and mesenchymal markers ($n \geq 3$) in agnospheres and tumorspheres mCRC729, mCRC1055 (from colorectal cancer metastases) and mMS321 (from melanoma metastasis).

Point N.3

Fig. 5B. AS67, and AS914 are missing from the blots. Moreover, other controls such as the colon cells used in “A” are missing for comparison purposes.

Reply

AS67 and AS914, together with other non-CUP tumorspheres have been added to former Fig.5B, now Fig5a (shown below).

Figure 5a. Representative Western blot analysis of phosphorylated (p) and total receptor tyrosine kinases, and phosphorylated MEK1/2 in agnospheres and tumorspheres ($n=3$).

Point N.4

Viability assays (CellTiter Glo), performed in Fig. 5C and other figures (e.g. S5D,E), are performed at too long of a timepoint. Viability is most often measured 72h after treatment. Here, the authors measured viability after 4-10 days. Using a prolonged treatment time complicates interpretation of the data, as media/drugs would have to be changed at least once over the course of the experiment, which can alter proliferation rates, and thus, the results. These experiments need to be repeated with shorter treatment times (72h).

Reply

The timepoints to perform viability assays have been carefully chosen based on: (i) the agnosphere duplication times (reported at page 9 of the original manuscript, and updated in the revised text for all agnospheres: 2 days for AS901, AS906 and AS67; 3.5 days for AS43, 5 days for AS914, and 10 days for AS47, page 11); (ii) the time required to detect a full apoptotic effect (if any), as established in preliminary time-course experiments. In particular, 10 days were required to observe a full apoptotic effect in AS43. As shown in the new Supplementary Fig. 5f (see below), in a dose-response experiment at 4 days, AS43 response to trametinib was incomplete, while, as shown in the original manuscript (Fig. 5C), after 10 days the response was complete. Please note that in the case of trametinib-resistant agnospheres, viability at the longest time-point was unchanged or paradoxically increased, indicating lack of unspecific toxic effects.

Supplementary Fig. 5. **f**, Cell viability measured in agnospheres treated for 4 days with the indicated doses of trametinib, selumetinib or vehicle (DMSO) through quantification of relative ATP consumption normalized vs. untreated cells (NT) (trametinib: AS901 n=4, AS906 and AS43 n=3, and selumetinib: AS901 n=4, AS906 and AS43 n=5, mean±SEM).

Regarding the Reviewer's suggestion (72 h time-point) and his/her concerns about medium change, please note that 72h is the time-point usually analyzed in conventional cell lines that are seeded at high density in FBS, replicate within hours and quickly exhaust the medium. In contrast, as described in the Methods section of the original manuscript (page 32, now page 37), during cell viability experiments, agnospheres are plated at low-density in a minimal medium devoid of any growth factor ('culture medium', described at page 23, now page 28-29), which does not require replacement for 10-14 days (as standardized for most CSC cultures). Finally, as trametinib half-life is longer than 4 days (Ref. 37), this drug needs to be added only at seeding and at day 5 (as reported in the Methods, page 37).

Point N.5

Fig. 5D. a) AS67 and AS914 are missing from the blots. This is particularly important since trametinib treatment has opposite effects in AS914 (Fig. S5G). The authors also indicate that the

paradoxical effects on AS914 are likely mediated by “rebound effect on upstream proliferative signals; pg. 15); however, mechanistic data to prove this hypothesis are never shown. b) pMEK blots are missing. This is critical given that the authors want to make the case that the MEK/ERK/MYC pathway is upregulated in these cultures, which is then targeted by trametinib. c) The blots (as well as those in Fig. S5F) were performed on lysates from cells treated for 48-96h. This timepoint is too long as it gives time for the cells to adapt and activate positive and negative feedback loops. The blots need to be performed at a short timepoint (<24h) to assess immediate drug responses. d) The authors demonstrate that trametinib treatment inhibits EGFR activation in two cell lines; however, they never identify the mechanism nor demonstrate that this downregulation (or expression of any RTK for that matter) is critical for trametinib sensitivity. The authors indicate that the mechanism of trametinib sensitivity may be tied to the finding that trametinib reduces EGFR activation, and thus, due to lack of ERK-mediated negative feedback regulation of EGFR. However, trametinib reduces pEGFR in both AS901 and AS43 cells (Fig. 5D) even though AS901 is insensitive to EGFR inhibition, whereas AS43 is dramatically sensitive to lapatinib/afatinib (Fig. S5D). Again, mechanistic experiments aimed at linking EGFR to MEK/ERK activation, trametinib sensitivity, and proliferation/survival are not complete.

Reply

(a) Blots showing the trametinib response in AS67 and AS914 have been added to the new Fig. 5c, as shown below. In the resistant AS914, trametinib, although capable of stably preventing ERK phosphorylation, does not cause the MYC downregulation observed in sensitive cases, indicating that AS914 proliferation is independent from ERK1/2. Moreover, a rebound effect on EGFR is not observed. We discuss these novel observations at page 19, discarding the hypothesis that AS914 resistance to trametinib is mediated by a rebound effect on EGFR.

Fig. 5c. Representative Western blot analysis of agnospheres treated 48 h (AS901, AS906 and AS67) or 96 h (AS43, AS914 and N-AS47) with vehicle (-) or trametinib 50 nM (+) (p-: phosphorylated; cl. PARP: cleaved PARP).

(b) In the revised Fig. 5B, now Fig. 5a (see above, point N.3) we added a Western blot showing MEK phosphorylation in all agnospheres and tumorspheres. Please note that in former Fig. 5D (now Fig. 5c, see above) we showed phosphorylation of the MEK1/2 direct target and downstream effector ERK1/2 as a reliable proxy of MEK phosphorylation.

(c) We added a Western blot after 30 min of trametinib treatment, showing ERK1/2 inhibition together with decreased pRB phosphorylation (Supplementary Fig 5g, shown below). However, we think that it is important to show the effects of trametinib on cell signaling at long time-points, just because this allows to investigate whether cells can adapt and resist to the drug: indeed only agnospheres that are sensitive to trametinib display such long-term downregulation of MYC, pRB and cyclin D. Conversely, agnospheres resistant to trametinib retain high levels of MYC, the master link between the MAP kinase pathway and cell cycle regulation.

Supplementary Fig. 5g. Representative western blot analysis of AS906 treated 30 min with selumetinib (500nM), trametinib (50nM) or vehicle (n=2).

(d) Concerning EGFR, we are sorry we generated a misunderstanding: we didn't mean to associate trametinib sensitivity with EGFR inhibition. Indeed, our first approach was to block agnosphere growth with pan-EGFR inhibition, which was however ineffective in AS901 (and in HRAS-mutated AS906, added to the revised manuscript, Supplementary Fig. 5c,e), or limited to proliferative arrest in AS43 (Supplementary Fig. 5D of the original manuscript). In Fig 5D (now Fig. 5c), we included EGFR analysis after trametinib treatment because upregulation of EGFR phosphorylation has been described as a feedback mechanism of resistance (Ref. 37). As expected, in sensitive agnospheres, we didn't observe EGFR rebound hyperactivation, but, in some agnospheres (AS901 and AS43), we observed a previously unreported EGFR downregulation. The mechanism of such downregulation remains unexplained, although we can hypothesize that it depends on MYC inhibition, as the EGFR promoter contains MYC/MAX binding sites. If data on EGFR are confusing (Fig. 5c of the revised manuscript, bottom lanes), we can remove them without subtracting any essential information.

Point N. 6

Fig. 5E,F. a) Tumor growth curves are plotted as "fold change". Since this removes critical information, the data instead should be shown as raw numbers (Tumor volume—mm³). b) In order to determine whether tumor growth is different between trametinib and vehicle-treatment mice, the growth rates should be analyzed with statistical methods such as linear mixed models, rather than performing statistics on each individual timepoint. c) The dose utilized in the animal experiment

(1mg/kg) is 2.5X higher than the RP2D dose and thus, the dose utilized in people (4.8mg/day vs 2mg/day), which calls into question whether the results are clinically relevant. d) The tumors need to be stained for pMEK/pERK/MYC in order to determine whether the pathway identified in vitro, also exists in vivo.

Reply

(a-b) In Fig. 5E (Fig. 5d in the revised manuscript) tumor growth curves are reported as ‘fold change’ as this method, although removing some information (starting tumor volume), allows to follow the dynamic tumor increase over time, normalized vs. the starting tumor volume. Please note that we performed unbiased tumor volume randomization at day -1 and that, within each mouse group, volume SEM was very small (please see Source Data file). Therefore, in Fig. 5, we maintained the tumor growth curve as volume fold changes, as this is more easily interpreted. However, as requested, statistics have been revised with two-way ANOVA to show overall statistical significance. Moreover, in the revised manuscript, we provided absolute tumor volume measurements for each mouse and the relative statistics with the linear mixed model as a supplementary information (Source Data file).

(c) Concerning the trametinib dose used in vivo, please see reply to General Comment, part v.

(d) pERK tumor immunohistochemistry has been added as a proxy for pMEK1/2 in the revised Fig. 5f (see above, reply to General Comment, part iii).

Point N. 7

Fig. 5G,H. The authors have a long discussion in the “Results” why they evaluated the effect of trametinib on IS tumor growth rather than on metastatic progression, indicating that the tumor growth mirrors metastatic growth. However, it is unclear why the authors didn’t evaluate the effect on metastatic growth at other organ sites, given that the cells are labeled and metastatic fluorescence can be evaluated. Since CUPs are highly metastatic, this seems to be an important evaluation for the effectiveness of trametinib. Moreover, in the discussion of Fig. 4, they indicate that organ metastases occurs early in the disease process, making it even more questionable why they didn’t assess trametinib’s effects at the ectopic sites.

Reply

As requested, in the revised manuscript, we added analysis of trametinib effectiveness on lung metastases (page 20 and Supplementary Fig. 5k, shown above, reply to General Comment, part vi). Concerning the above Reviewer’s observations, we’d like to point out that the experiment shown in Fig. 5E-F of the original manuscript (now Fig. 5d-e) was planned to investigate the trametinib effect on subcutaneous tumors, and thus it was performed with unlabeled wild-type cells (as described in

the Methods section and further clarified at page 40 of the revised manuscript). In this context, in vivo bioluminescent imaging was obviously impossible. However, we could analyze lungs collected in mice transplanted with AS43 at the endpoint (1600 m³) of the experiment shown in Fig. 5d-e, showing that trametinib decreases the number and size of metastatic emboli (Supplementary Fig. 5k). Unfortunately, within the time frame of manuscript revision, we could not perform a new dedicated experiment with labelled agnospheres, as suggested. Indeed, given the range of organ colonized (Fig. 4f), and the variable size of metastases in different animals, to set up statistically significant experiments a large number of mice is needed. This requires a long approval procedure by the competent authority (Italian Ministry of Health).

In further support of the possible trametinib effectiveness on multiple metastases, as described above (see Reply to General Comment, part vi), we added a new analysis showing that metastases collected at warm autopsy from the human patient AGN43 (the origin of AS43) display the same trametinib response signature (new Fig. 5i, shown above).

Point N.8

Fig. 5I,J. The authors show that a 1,000 gene signature can predict sensitivity to trametinib; however, they never confirm the data by other means (e.g. western blot), and never use information from this signature to perform mechanistic (rescue) experiments aimed at identifying critical proteins whose expression can reverse trametinib sensitivity in agrosphere cultures.

Reply

RNAseq data integrated with proteomics data are currently under scrutiny to identify molecular mechanisms of sensitivity or resistance, which are very complex (involving not only transcriptional but also post-translation modulation), and whose disclosure seems beyond the scope of this manuscript.

Point N. 9

Fig. S5C. Data for AS914 are missing.

Reply

In the revised manuscript, IHC for EGFR and pEGFR, shown in Fig. S5C of the original manuscript, has been replaced by Western blots including AS914 (see new Fig. 5a, shown above, Reply to Point N.3).

Point N. 10

Fig. S5D. Cell viability data for AS901, AS906, AS914 are missing even though the authors make the point that only one agnosphere responded to the drugs, indicating that they tested all of the cell lines. Also, the cell viability assays were performed at too long of timepoints (see point #5).

Reply

We attempted EGFR pan-inhibition in representative agnospheres AS43 and AS901, shown in Supplementary Fig. 5D-E of the original manuscript. AS906 and AS914 are not supposed to respond to EGFR inhibition, as they harbor HRAS and KRAS mutations, respectively, well-known determinants of primary resistance to EGFR therapy. However, as further confirmation, in the revised text we added cell viability and Western blot analysis in AS906 in response to pan-EGFR inhibitors (new Supplementary Fig 5c-e, see below).

Concerning the cell viability timepoints, they were established based on criteria (cell population doubling time, time required to detect a full effect) described in the Reply to Point N.4 (see above).

Supplementary Fig. 5c-e. **c**, Agnosphere viability evaluated 4 days after treatment with the indicated EGFR inhibitors, lapatinib (0.5 μm) or afatinib (0.5 μm), or vehicle (DMSO), through quantification of relative ATP consumption normalized vs. untreated condition (NT) ($n \geq 3$, mean \pm SEM, $*p < 0.05$, ANOVA, Bonferroni multicomparison test: Lapatinib vs. DMSO and Afatinib vs. DMSO). **d**, AS43 viability evaluated at the indicated time-points after treatment with the indicated EGFR inhibitors or vehicle (DMSO) through quantification of relative ATP consumption normalized vs. untreated condition (NT) at day 4 ($n \geq 3$, mean \pm SEM, ns: not significant, ANOVA, Bonferroni multicomparison test). **e**, Representative western blot analysis of agnospheres treated 4 days with the indicated inhibitors. pAKT: phospho-AKT; pERK1/2: phospho-ERK1/2; p-p38: phospho-p38-MAPK; pSTAT1: phospho-STAT1; pSTAT3: phospho-STAT3; p27: p27-KIP1; p-pRB: phospho-RB. Actin B was used as loading control.

Point N.11

Fig. S5F. The timepoint utilized for preparing the lysates for western blots is too long (see point #5c above), and blots for the other cell lines are missing (only two agnospheres are shown). In the text (pg. 14), the authors indicate that the ERBB inhibitors are likely ineffective in preventing viability in the majority of lines (except for AS43) because the cells have found another way to

activate EGFR signaling (pERK, pAKT). However, neither ERK nor AKT phosphorylation (EGFR downstream signaling) is altered in either AS43 (sensitive) or AS901 (resistant) cells following lapatinib/afatinib treatment, indicating that this isn't a likely explanation. Critical mechanistic experiments aimed at uncovering why ERBB inhibitors are effective (or ineffective) are lacking. Also, the afatinib label is missing from the figure.

Reply

As agnospheres display multiple concomitant growth factor autocrine loops (and, in some cases, RAS mutations), it seems conceivable that targeting only one receptor, although prominent like EGFR, is ineffective or only partly effective. Based on Western blots results (former Supplementary Fig. S5F, now Supplementary Fig. 5e, shown above, Reply to point N.10), EGFR phosphorylation is fully prevented by inhibitors, but the entire downstream pathways are intact. Thus, a likely explanation is that some bypass signaling (possibly elicited by another tyrosine kinase receptor) keeps MAP kinase and AKT pathways active. However, given lack of biological effectiveness of EGFR inhibitors, it seems beyond the scope of this study to investigate such bypass mechanisms.

Point N. 12

Fig. S5G. Trametinib viability data is only shown for two agnospheres—there rest are missing.

Reply

In the original manuscript, data on agnosphere treatment with trametinib were complete but split between the main Fig. 5C and Supplementary Fig. 5G. In the revised manuscript, to prevent any misunderstanding, we displayed data on all agnospheres in the same panel (Fig. 5b, see below).

Figure 5b. Cell viability of agnospheres treated for 4 days (AS901 and AS906) or 10 days with the indicated doses of trametinib (50 nM) or vehicle (DMSO) normalized vs. untreated cells (NT) (AS901 n=4, AS906 n=3, AS43 and N-AS47 n=5, mean±SEM, ANOVA, Bonferroni multicomparison test, ***p ≤ 0.001).

Reviewers' Comments:

Reviewer #1:

Remarks to the Author:

Major comments:

(1) The authors have now added two colospheres, derived from metastases of colorectal cancers and one melanosphere, derived from a melanoma metastasis as controls to some of the experiments performed. As CUP, at least according to most autopsy studies, is mostly derived from pancreatic cancers and non-small cell lung cancers, the addition of controls derived from those cancer types would have been preferable. Also, melanoma is not an epithelial cancer type and therefore not well suited for comparison.

(2) An important point of Figure 2d - the lack of a difference between agnospheres and tumorspheres - should be stated more clearly.

(3) Figure 2e is not mentioned in the text and no data on tumorspheres is presented for comparison. Similarly, although the respective numbers are given from the literature, no own data is presented for tumorspheres in in vivo limiting dilution assays to determine TIC frequencies (Figure 3) for comparison either. Also, experiments on the in vivo metastatic capacity of agnospheres (Figure 4) include only one colosphere control. Why have those experiments not at least been performed with all available control tumorspheres (2 colospheres, 1 melanosphere)?

The authors state in the title of the manuscript that "stem-like cells from cancer of unknown primary are endowed with distinctive hypermetastatic properties ...". Similarly, it is stated in the abstract that CUPs "display exceptionally high tumorigenicity". To allow for these conclusions, proper controls are required.

Minor comment:

As in the current study only a few cases have been analysed and CUP-specific mutational spectra do not exist, I would suggest to indeed leave out this point as proposed by the authors in their rebuttal letter.

Reviewer #2:

Remarks to the Author:

Major points:

1. Addressed, they included controls. Satisfied with new Fig 2 overall. A concern remains for Fig 2A as it is not appropriate to show mRNA quantification in the heatmap using CT values.
2. Addressed.
3. Concern: while the authors show number of cells per metastatic embolus in Suppl Fig 5K this is only of one slice so the units need to be redefined or show area of metastatic nodules per section?
4. Addressed.

Minor points:

1. Addressed.
2. Addressed
3. Addressed in words.
4. Addressed not with genetic proof but with other MEK inhibitor.
5. Addressed.
6. Addressed.

Reviewer #3:

Remarks to the Author:

This reviewer is satisfied with the rebuttal comments, and/or changes/additions made to the manuscript in response to my criticisms.

NCOMMS-20-28191A

“Stem-like cells from Cancer of Unknown Primary (CUP) are endowed with distinctive hypermetastatic properties and unveil liability to MEK inhibition” by Federica Verginelli et al.

POINT-BY-POINT REPLY TO REVIEWERS' COMMENTS

Reviewer #1

We thank this Reviewer for the overall positive evaluation of the revised manuscript.

Major comment N.1

The authors have now added two colospheres, derived from metastases of colorectal cancers and one melanosphere, derived from a melanoma metastasis as controls to some of the experiments performed. As CUP, at least according to most autopsy studies, is mostly derived from pancreatic cancers and non-small cell lung cancers, the addition of controls derived from those cancer types would have been preferable. Also, melanoma is not an epithelial cancer type and therefore not well suited for comparison.

Reply

We agree that extending the comparisons among CUPs and other metastatic cancers would add valuable information. However please take in account that adding the proposed models would imply: (i) deriving tumorspheres from metastases of pancreatic and lung cancers, which are seldom amenable to biopsy if the primary tumor is recognized, and would require a dedicated, ethically approved study; (ii) finding tumorspheres with the ability to self-sustain long-term propagation like agnospheres, which could require many attempts; such an investigation is hardly feasible within the scope of the current manuscript. Concerning tumorspheres derived from metastatic melanoma (melanosphere mMS321), we think this is a valuable comparison as it represents the tumor most enriched in tumor-initiating cells ever measured before (Ishizawa et al., Cell Stem Cell 7:279, 2010; Quintana et al., Nature 456:593, 2008). Showing that agnospheres contain a fraction of tumor-initiating cells comparable to that of metastatic melanoma (Fig. 3c and Supplementary Fig. 3c, now Supplementary Fig. 2c) highlights the extreme tumorigenicity of such epithelial, but overall undifferentiated, tumors. Finally, please note that melanosphere mMS321 harbors BRAFV600E mutation, known to confer sensitivity to trametinib.

Therefore, this melanosphere represents the ideal positive control for experiments where we measure the agnosphere response to trametinib.

Major comment N.2

An important point of Figure 2d - the lack of a difference between agnospheres and tumorspheres - should be stated more clearly.

Reply

Lack of difference between agnospheres and tumorspheres in Figure 2d has been better highlighted at page 9: ‘By direct comparison, agnospheres and tumorspheres displayed global levels of Polycomb and MYC target genes overall similar and comparable to those of ES cells (...)’.

Major comment N.3

Figure 2e is not mentioned in the text and no data on tumorspheres is presented for comparison. Similarly, although the respective numbers are given from the literature, no own data is presented for tumorspheres in in vivo limiting dilution assays to determine TIC frequencies (Figure 3) for comparison either. Also, experiments on the in vivo metastatic capacity of agnospheres (Figure 4) include only one colosphere control. Why have those experiments not at least been performed with all available control tumorspheres (2 colospheres, 1 melanosphere)?

The authors state in the title of the manuscript that "stem-like cells from cancer of unknown primary are endowed with distinctive hypermetastatic properties ...". Similarly, it is stated in the abstract that CUPs "display exceptionally high tumorigenicity". To allow for these conclusions, proper controls are required.

Reply

- (i) We apologize for involuntarily omitting citation of Fig. 2e, which has now been added (page 11).
- (ii) Concerning TIC frequency evaluation, please note that indeed we added our own TIC measurement in melanosphere mMS321 (Supplementary Fig. 3c, now Supplementary Fig. 2c), as described in the main text, page 13. As stated above (reply to major comment N. 1) and reported in the text, we think this is a particularly important control, as melanoma is known as the tumor containing the highest TIC frequency.
- (iii) Concerning evaluation of metastatic ability, to demonstrate lack of metastases by colospheres mCRC729 required an overall 7-months *in vivo* monitoring. Repeating this kind of experiment with

other available colospheres would therefore require an amount of time that seems hardly justified for the assessment of additional, hypothetical negative controls. Concerning melanomas, as stated at page 15 of the revised manuscript, it is known that, unlike carcinomas, they can metastasize when injected *subcutis*, which can be considered an orthotopic growth site.

(iv) We understand the concerns about possible overstatements of in the title and abstract. Therefore we propose to change the title as follows: ‘Cancers of Unknown Primary stem-like cells model multi-organ metastasis and unveil liability to MEK inhibition’. In the abstract, we deleted the word ‘exceptionally’ from the statement ‘displaying exceptionally high tumorigenicity’. Moreover, we removed emphatic terms concerning agnosphere properties throughout the text.

Minor comment

As in the current study only a few cases have been analysed and CUP-specific mutational spectra do not exist, I would suggest to indeed leave out this point as proposed by the authors in their rebuttal letter.

Reply

As proposed, we deleted the statement about CUP mutations from Results, page 7, and we removed the related Supplementary Fig. 1.

Reviewer #2

We thank this Reviewer for the mostly positive evaluation of the revised manuscript.

Major points

1. Addressed, they included controls. Satisfied with new Fig 2 overall. A concern remains for Fig 2A as it is not appropriate to show mRNA quantification in the heatmap using CT values.

Reply

As reported in the legend of original Fig. 2A, expression (Ct value) of each gene was normalized vs. HPRT expression (housekeeper). This has been now clarified in the panel labelling. This normalization should prevent any concern about the significance of the relative gene expressions shown in this figure as well as in previous Supplementary Fig. 5d (now Supplementary Fig. 4d, whose labelling has been updated consistently).

2. *Addressed.*

3. *Concern: while the authors show number of cells per metastatic embolus in Suppl Fig 5K this is only of one slice so the units need to be redefined or show area of metastatic nodules per section?*

Reply

We apologize for the unclear description of metastatic emboli counting. In former Supplementary Fig. 5k (now Supplementary Fig. 4l) we reported the total number of emboli counted in all lung sections from every mouse, and not in a single slice (the H&E image shown in the current Supplementary Fig. 4k is only representative of emboli appearance and dimension). As specified in Supplementary Methods (page 25), emboli were counted in sections spaced 20 μm to avoid duplicating counts of the same cells. We have now revised the Figure legend to clarify that emboli were counted in all lung sections.

4. *Addressed.*

Minor points

1. *Addressed.*

2. *Addressed*

3. *Addressed in words.*

4. *Addressed not with genetic proof but with other MEK inhibitor.*

5. *Addressed.*

6. *Addressed.*

Reviewer #3

General comment

This reviewer is satisfied with the rebuttal comments, and/or changes/additions made to the manuscript in response to my criticisms.

Reply. We thank this Reviewer for the fully positive evaluation of the revised manuscript.